# Rational design of a trispecific antibody targeting the HIV-1 Env with elevated anti-viral activity

James J. Steinhardt[1,2], Javier Guenaga[3], Hannah L. Turner[4], Krisha McKee[5], Mark K. Louder [5], Sijy O'Dell[5], Chi-I Chiang[1], Lin Lei[1], Andrey Galkin[1,6], Alexander K. Andrianov[1], Nicole A. Doria-Rose[5], Robert T. Bailer[5], Andrew B. Ward[4], John R. Mascola[5] & Yuxing Li[1,2,6]

HIV-1 broadly neutralizing antibodies (bNAbs) are being explored as passively administered therapeutic and preventative agents. However, the extensively diversified HIV-1 envelope glycoproteins (Env) rapidly acquire mutations to evade individual bNAbs in monotherapy regimens. The use of a "single" agent to simultaneously target distinct Env epitopes is desirable to overcome viral diversity. Here, we report the use of tandem single-chain variable fragment (ScFv) domains of two bNAbs, specific for the CD4-binding site and V3 glycan patch, to form anti-HIV-1 bispecific ScFvs (Bi-ScFvs). The optimal Bi-ScFv crosslinks adjacent protomers within one HIV-1 Env spike and has greater neutralization breadth than its parental bNAbs. Furthermore, the combination of this Bi-ScFv with a third bNAb recognizing the Env membrane proximal external region (MPER) results in a trispecific bNAb, which has nearly pan-isolate neutralization breadth and high potency. Thus, multispecific antibodies combining functional moieties of bNAbs could achieve outstanding neutralization capacity with augmented avidity.

[1] Institute for Bioscience and Biotechnology Research, University of Maryland, Rockville, MD 20850, USA. [2] Virology Program at the University of Maryland, College Park, MD 20740, USA. [3] IAVI Neutralizing Antibody Center, The Scripps Research Institute, La Jolla, CA 92037, USA. [4] Department of Integrative Structural and Computational Biology, The Scripps Research Institute, La Jolla, CA 92037, USA. [5] Vaccine Research Center, National Institute of Allergy and Infectious Diseases, National Institutes of Health, Bethesda, MD 20892, USA. [6] Department of Microbiology and Immunology, University of Maryland School of Medicine, Baltimore, MD 21201, USA. Correspondence and requests for materials should be addressed to Y.L. (email: liy@ibbr.umd.edu)

Recent advances in the discovery of broadly neutralizing antibodies (bNAbs) targeting the HIV-1 envelope glycoproteins (Env) have awakened great interest in their use as pre-exposure prophylaxis for prevention and as therapeutic agents, particularly in combination with antiretroviral treatment (ART) for HIV remission and eradication[1–3]. bNAb isolation and characterization has been accelerated via the integration of emerging functional and structural information and new technologies of single B cell sorting and cloning[4–9]. bNAbs are therapeutically beneficial as they possess high capacity for viral neutralization. Additionally, bNAbs can facilitate fragment crystallizable (Fc)-mediated effector functions that promote cell lysis and/or clearance of infected cells that express HIV-1 Env on the cell surface via antibody-dependent cell-mediated cytotoxicity and complement-dependent cytotoxicity[10].

The characterization of HIV-1 bNAbs and their cognate epitopes on the Env spikes has identified five conserved Env sites of vulnerability including the CD4-binding site (CD4bs), the V1/V2-glycan region, the V3-glycan region, the gp41 membrane proximal external region (MPER), and the gp120–gp41 interface[11]. Passive immunization with bNAbs is being explored as a means for prevention in healthy individuals and as treatment for HIV infected patients. Passive immunization in humans has proven highly effective in treating many infections such as hepatitis A, hepatitis B, rabies, and respiratory syncytial viruses[12], but these viruses have much lower genetic diversity[12–16] than do circulating HIV-1 isolates[17, 18], which greatly confounds the clinical outcome of passive immunization for HIV-1 treatment. Administration of a single bNAb as a therapeutic agent has successfully cleared phase I safety clinical trials, demonstrating temporary HIV-1 viremia suppression in the majority of patients[19, 20]. Unfortunately, the HIV virus rapidly develops resistance mutations under pressure from a single bNAb, suggesting that passive treatment with a single bNAb is unlikely to result in long-term viremia suppression[19, 21–23]. Some of the Env mutations associated with bNAb resistance can significantly reduce viral fitness. Therefore, simultaneously targeting different Env epitopes may completely compromise viral replication, as mutations that confer resistance to each bNAb often accumulate to severely reduce viral fitness[24–27]. In addition, treatment of simian/human immunodeficiency virus infection in non-human primate models demonstrated that passive immunotherapy with bNAb cocktails prevent mother to child transmission, suppress viremia and, in contrast to combinatorial antiviral therapy (cART) treatments, facilitate CD8+ T-cell immunity for durable suppression of virus replication[28, 29]. Preliminary data on bNAb cocktails suggest significant advantages over either cART or single bNAb treatments for the management of HIV-1 infection.

While antibody cocktails demonstrated improved efficacy in preclinical studies, multispecific "single agents" are desirable for manufacturing purposes[30] as well as for improved avidity that may result in enhanced neutralization breadth and potency[31]. Bi-NAbs with two Env-epitope binding sites have been generated using CrossMab formats[32, 33] with up to 97% virus coverage[34, 35]. Their neutralization breadth could be further extended, however, and truly bivalent binding has yet to be experimentally demonstrated. Most recently, one study showed that swapping the IgG1 hinge for a more flexibly IgG3 hinge lacking disulfide bonds (denoted as IgG3C-) greatly improved the potency of anti-HIV CrossMabs[34]. While both the CrossMab and IgG3C- designs have significantly improved the potency and breadth of antibodies against HIV, they only target two epitopes, one corresponding to each antigen-binding Fragment (Fab) arm. This limits the potential increase of avidity that would result from simultaneous engagement of multiple functional moieties. Furthermore, the traditional CrossMab format imposes steric constraints that may impede true bivalent engagement of the Fab arms due to the rigidity of the dimeric IgG Fc fragment where the Fabs are placed[31].

One study has demonstrated the use of DNA-linkers as "molecular rulers" to connect Fab moieties of two bNAbs resulting in molecules capable of intra-spike crosslinking to enhance the avidity and potency of bNAbs[31]. The chemical conjugation process required for connecting Fabs with DNA-linker[31] in this method, however, limits its feasibility and application scale. We therefore sought to expand the concept of simultaneous engagement of multiple epitopes within a spike by generating antibodies using Bi-ScFvs simply joined together by $(G_4S)_n$ flexible linkers[36, 37]. ScFvs joined by linkers have been manufactured for preclinical studies previously[38]. We made use of two prototypical bNAbs, VRC01, targeting CD4bs, the Env receptor-binding site[8], and PGT121, targeting the N332 glycan of the V3 region[9], as a Bi-ScFv model. On the basis of Env and bNAb high-resolution structural information, we selected the combination of bNAbs and determined the length of the $G_4S$ linkers that would facilitate intra-spike crosslinking of the Env. Subsequently, we fused these ScFvs to the IgG Fc to accommodate effector functions[39], and we paired our best bispecific antibody with an MPER-specific antibody 10E8 to generate a Tri-NAb. We show here that the Tri-NAb achieved near-pan virus neutralization breadth (99.5% of virus coverage) tested in a comprehensive 208 virus panel with $IC_{50}$ geometric mean below 0.1 μg/mL, greatly improving upon the capacities of the individual parental bNAbs. This multispecific NAb can be of great interest in future studies aimed at preventing disease progression or mother to child transmission and curing HIV. Furthermore, our approach that combines multi-functional moieties of individual bNAbs with profoundly elevated avidity and cooperative effect of multivalence interactions may be applied to generate superior antibody-based antiviral therapeutics against other infectious agents.

## Results

**Structural analysis of the HIV-1 Env trimer-bNAb complexes.** For the first step towards generating multi-epitope targeting bNAbs, we sought to engineer bispecific single-chain variable fragment (Bi-ScFv). This construct consists of two bNAb variable fragments (Fv) connected by a flexible amino acid linker whose length was estimated following Env/bNAb structural information[36, 37]. We selected bNAbs VRC01 (CD4bs-directed)[8] and PGT121 (V3-base glycan-directed)[9], which neutralize ~90 and 70% of circulating viruses, respectively, as a model system to determine whether a Bi-ScFv could improve breadth and potency over the capacities of the individual bNAbs. To guide our design, we used a bNAb/Env complex structure[40] to assess the distance between the bNAbs Fvs and to determine ideal linker lengths[41].

To determine the distances between the variable domains of VCR01 and PGT121, we used the PDB structure 5FYK[40], which includes the HIV-1 clade B trimeric Env JR-FL SOSIP.664 bound to bNAbs VRC01 and PGT122. PGT122 is a somatic variant of PGT121 possessing nearly identical Env binding mode[9] that serves as a surrogate for PGT121 in this study (Fig. 1a). Utilizing the distance tool under the structure analysis module in UCSF Chimera[41], we examined the distances between the C/N -termini of VRC01 and PGT121 VH/VL moieties for all eight termini combinations (Supplementary Table 1). We found that in general the distance between these two antibody moieties on two separate but adjacent gp120 protomers (inter-protomer distance) is shorter than that within the same protomer (intra-protomer distance) (Fig. 1a and Supplementary Table 1). While the minimum intra-protomer distance between PGT121 and

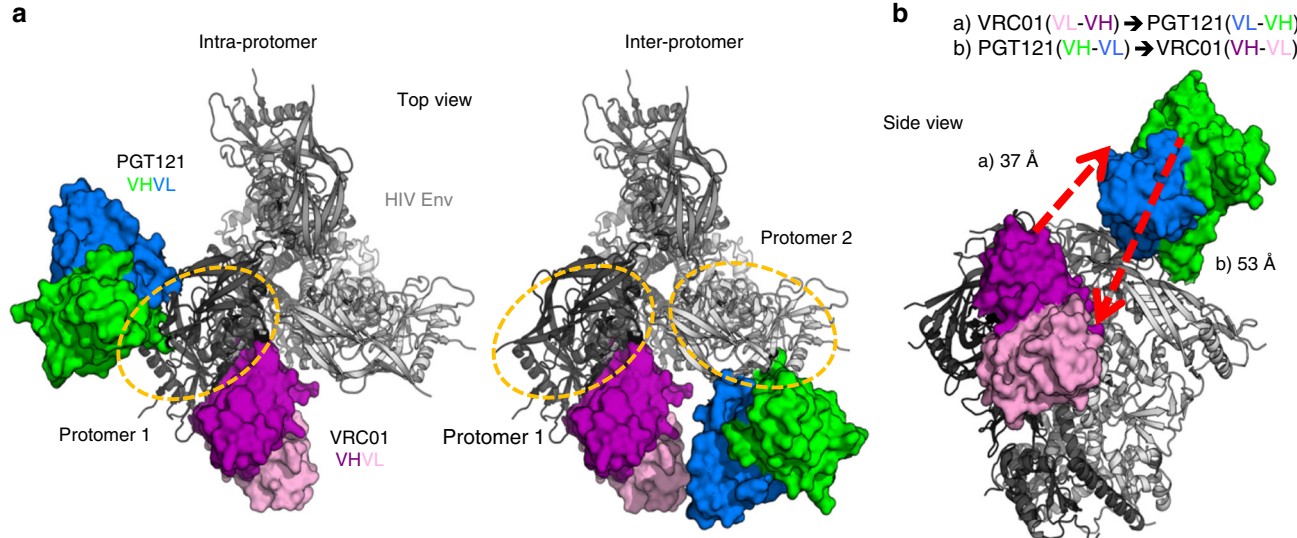

**Fig. 1** Design of bispecific antibodies. **a** Structure of HIV-1 JR-FL SOSIP.664 Env trimer (PDB: 5FYK) showing the footprints of bNAbs VRC01 and PGT121 and their proximity in both intra-protomer (left) and inter-protomer (right) binding configurations. PGT122 serves as a surrogate for PGT121. **b** Distances between VRC01 and PGT121 VH/VL termini and two Bi-ScFv molecules of different topology

VRC01 termini is ~75 Å, the minimum inter-protomer distance is 37.2 Å (VRC01 $VH_{C-term}$ to PGT121 $VL_{N-term}$) and 53.5 Å (PGT121 $VL_{C-term}$ to VRC01 $VH_{N-term}$), respectively (Fig. 1b and Supplementary Table 1). The relatively short VRC01 and PGT121 VH/VL N/C termini inter-protomer distance implies the potential to connect their antibody functional moieties by tandem GGGGS ($G_4S$) linkers without unfavorable steric clash and supports the premise of inter-protomer VRC01/PGT121 Bi-ScFvs to target HIV-1 Env trimer.

**Design and expression of bispecific antibodies**. With the rationale stated above, we selected the VRC01/PGT121 inter-protomer binding mode (Fig. 1b) and the most favorable molecular topology, including $VRC01_{VL-VH} \rightarrow PGT121_{VL-VH}$ and $PGT121_{VH-VL} \rightarrow VRC01_{VH-VL}$ respectively, to construct Bi-ScFvs (Fig. 1a). First, to derive the individual ScFvs, we used three tandem $G_4S$ linkers (designated as 3X linker)[42, 43], with each linker estimated to be 18 Å in length, to connect the cognate VH/VL domains within VRC01 and PGT121, respectively (Fig. 2a). We then used 3–5 (3–5X) $G_4S$ linkers, to connect the individual VRC01 and PGT121 ScFvs to form the Bi-ScFv molecule (Fig. 2a). The empirical choice of 3–5 $G_4S$ linkers was to avoid potential steric hindrance imposed by elements of the HIV-1 Env functional spike. Hence, the nomenclatures of these Bi-ScFv constructs, shown in Fig. 2a included: (1) the molecular topology of the whole Bi-ScFv (e.g., VRC01 ScFv at the N-terminus), and (2) the linker length between each ScFv (e.g. 3 tandem $G_4S$ linker). For instance, $Bi-ScFv_{VRC01-5X-PGT121}$ represents a Bi-ScFv with $VRC01_{VL-VH} \rightarrow (G_4S)_5$ linkers $\rightarrow PGT121_{VL-VH}$ topology, while $Bi-ScFv_{PGT121-5X-VRC01}$ ScFv denotes a Bi-ScFv with $PGT121_{VH-VL} \rightarrow (G_4S)_5$ linkers $\rightarrow VRC01_{VH-VL}$ topology (Fig. 2a).

In a previous study, a truncation of the first two amino acid residues E1 and I2 (ΔE1I2) and a V3S mutation at the N-terminus of VRC01 VL increased potency of HIV neutralization[44] by eliminating a steric clash between the VRC01 VL N-terminus and the V5 region of HIV-1 Env. Therefore, we incorporated such modifications into the VRC01 VL moiety of the $Bi-ScFv_{VRC01-5X-PGT121}$, denoted as $Bi-ScFv_{dVRC01-5X-PGT121}$ (Fig. 2a). These ScFvs were also fused to the IgG1 Fc via a glycine-serine-glycine linker to support effector functions[39], thus forming Bi-NAbs (Fig. 2a, b).

Ten bispecific antibodies of the PGT121/VRC01 Bi-ScFvs and Bi-NAbs iterations in total were expressed in 293 F cells and purified by affinity chromatography. All the antibodies expressed well and ran as homogeneous species at the expected molecular weight on SDS–PAGE gels (Supplementary Fig. 1A & B).

**Binding properties of bispecific antibodies**. We used BioLayer Interferometry (BLI) to validate the simultaneous Env trimer engagement of the two arms of the Bi-ScFvs. The Env ligands include RSC3 core, which selectively displays the CD4bs (VRC01) epitope but not the PGT1210 epitope, and the BG505 SOSIP.664_D368R mutant that exclusively exhibits the PGT121 epitope (Fig. 3a). In this assay, we first loaded biotinylated RSC3 as the initial ligand (ligand 1), then the parental or bispecific antibodies, followed by the second ligand BG505 SOSIP.664_D368R (ligand 2; Fig. 3a), and assessed the antibody binding signals to each of the Env ligands. As expected, we observed that only antibodies containing the VRC01 moiety including $Bi-ScFv_{dVRC01-5X-PGT121}$ and $Bi-NAb_{dVRC01-5X-PGT121}$ as well as VRC01 Fab and IgG, could bind CD4bs ligand, RSC3 (Ligand 1) initially (Fig. 3b). Similarly, bispecific antibodies containing the PGT121 moiety bound the CD4bs knockout trimer (VRC01-KO), BG505 SOSIP.664_D368R, which exclusively presents the PGT121 epitope, while the VRC01 Fab or IgG displayed no binding signal (Fig. 3b). These data confirm that bispecific antibodies including $Bi-ScFv_{dVRC01-5X-PGT121}$ and $Bi-NAb_{dVRC01-5X-PGT121}$ can simultaneously bind both the VRC01 and the PGT121 epitopes, demonstrating that both arms of the bispecific antibody are functional.

As shown in Fig. 1, we designed the bispecific antibodies to cross-link adjacent protomers within the same HIV-1 Env trimer by connecting the CD4bs of one Env protomer (protomer 1) with the glycan patch at the V3 base of the adjacent protomer (protomer 2). To further investigate the bivalency of $Bi-ScFv_{dVRC01-5X-PGT121}$, we performed negative stain electron microscopy (EM) analysis of the $Bi-ScFv_{dVRC01-5X-PGT121}$ and Env trimer BG505 SOSIP.664 complex (Fig. 3c, Supplementary Fig. 2A-B). We incubated the $Bi-ScFv_{dVRC01-5X-PGT121}$ with the BG505 SOSIP.664 Env trimer at a 1:2 (or 0.5:1) ratio so that each Env trimer would be occupied by a single $Bi-ScFv_{dVRC01-5X-PGT121}$

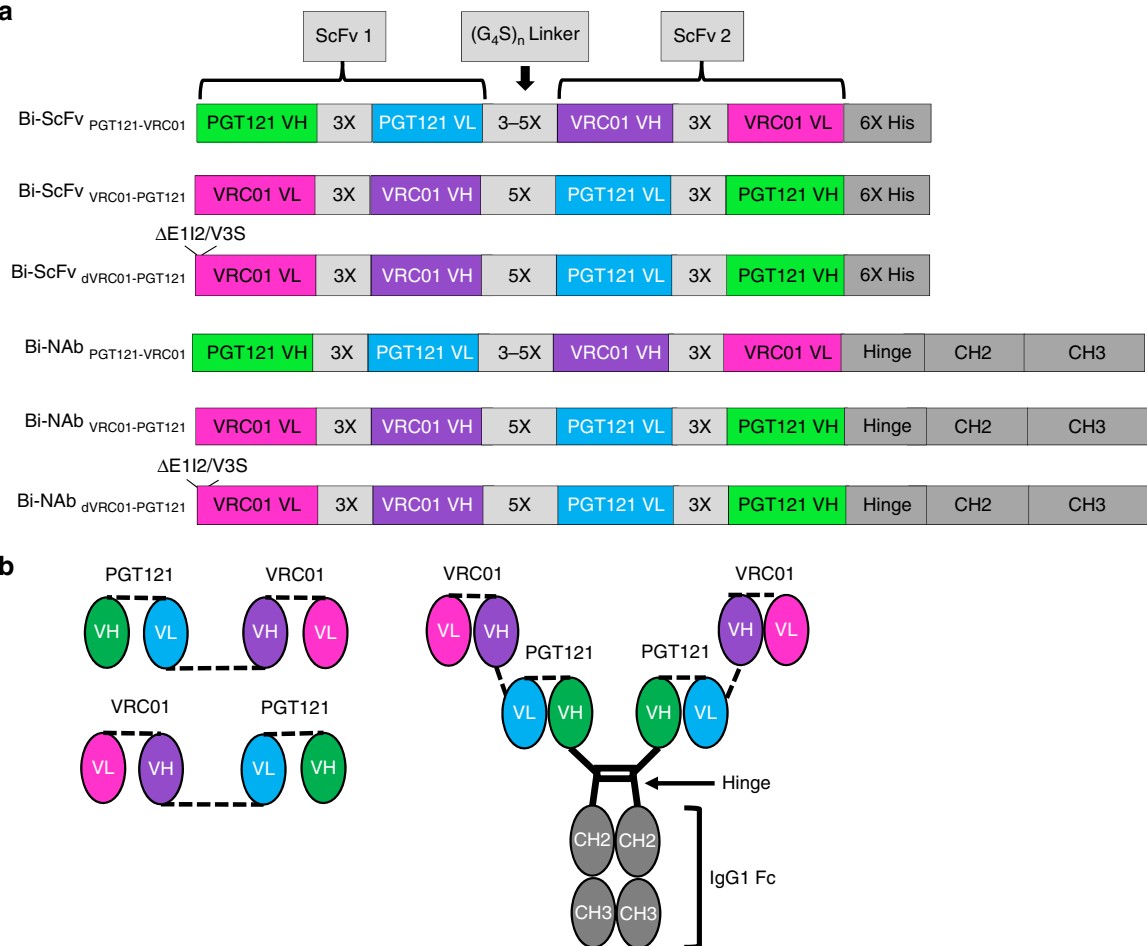

**Fig. 2** Bispecific antibody constructs. **a** Schematic presentation of the Bi-ScFv and Bi-NAb antibody constructs. **b** Schematic diagram of the molecular configurations of the Bi-ScFv and Bi-NAb antibodies

molecule, in a way that enabled us to distinguish the inter-protomer vs. intra-protomer binding modes depicted in Fig. 1a. As expected, we visualized both unbound Env trimer (Fig. 3c, left panel) and Env trimer bound with single Bi-ScFv$_{dVRC01-5X-PGT121}$ (Fig. 3c, second from left and middle panel), given the sub-saturation ratio of Bi-ScFv and Env trimer. We found that each Env protomer was decorated with only one of the Bi-ScFv moieties (Fig. 3c, second from left and middle panel), while two adjacent Env protomers were simultaneously decorated by two Bi-ScFv moieties (Fig. 3c, second from left and middle panel). The data corroborated the bivalent nature of the binding event between the Bi-ScFv and Env trimer, as predicted by design. The Bi-ScFv crosslinked both the VRC01 (in purple, Fig. 3c, middle panel) and PGT121 (in green, Fig. 3c, middle panel) epitopes in an inter-protomer mode. When we increased the molar ratio of Bi-ScFv$_{dVRC01-5X-PGT121}$ to BG505 SOSIP.664 trimer to 6:1 to form Bi-ScFv/Env complex, we observed that all the protomers of Env trimer were fully occupied with Bi-ScFv moieties and each Env trimer bound three Bi-ScFvs (Fig. 3c, second from right and right panel). This observation is consistent with the inter-protomer crosslinking binding mode and suggests that three molecules of Bi-ScFv$_{dVRC01-5X-PGT121}$ can fully occupy the total of six cognate epitopes on each HIV-1 Env trimer.

**Expanded neutralization breadth of bispecific antibodies.** To initially assess the virus neutralization capacity of the bispecific antibodies in comparison to their parental bNAbs, VRC01 and

PGT121, we selected a small HIV-1 virus panel ($N = 20$) containing Envs of viruses from diverse clades to perform virus neutralization assay (Supplementary Fig. 3A). This panel included Envs of viruses that were: (1) sensitive to both (dual sensitive) bNAbs ($N = 6$); or (2) resistant to both (dual resistant) bNAbs ($N = 1$); or (3) sensitive to one but resistant to the other parental antibody ($N = 13$; Supplementary Fig. 3A). It is notable that 70% of the selected viruses ($N = 14$) were resistant to at least one of the parental bNAbs, which represented a high bar for the evaluation of neutralization capacity.

Using IC$_{50}$ titers (the concentration of antibody at which 50% of virus entry is inhibited) with a cutoff value set at 50 μg/mL, we assessed the neutralization capacity of bispecific antibodies with this initial virus panel. We found that all the bispecific antibodies displayed substantially improved neutralization breadth, ranging from 80–90% virus coverage, compared to 60–65% virus coverage of their parental bNAbs (Fig. 4a, Supplementary Fig. 3A-B). Interestingly, we observed that the bispecific antibodies with the topology of VRC01$_{VL-VH}$ → PGT121$_{VL-VH}$ displayed 90% neutralization breadth, which was better than that with the PGT121$_{VH-VL}$ → VRC01$_{VH-VL}$ topology (80–85%; Fig. 4a). We also noted that the five-or four-tandem G$_4$S linker length (5X or 4X) was slightly better than the shorter variants (3X; Fig. 4a, Supplementary Fig. 3A-B) in many cases. Finally, the bispecific antibodies, Bi-ScFv$_{dVRC01-5X-PGT121}$ and Bi-NAb$_{dVRC01-5X-PGT121}$ with the optimal VRC01$_{VL-VH}$ → PGT121$_{VL-VH}$ topology, the 5X G$_4$S linker and the N-terminus VRC01 VL modifications

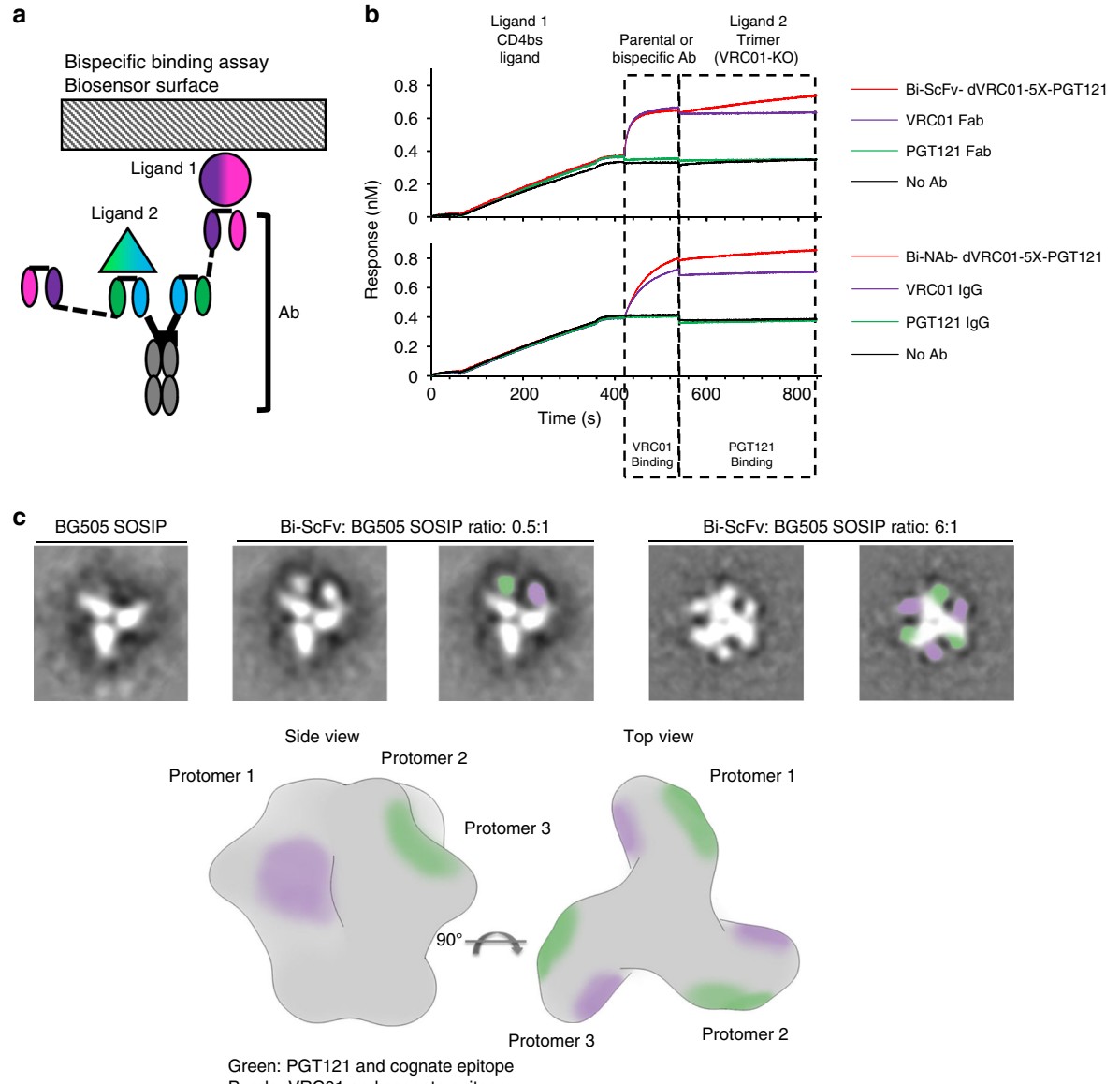

**Fig. 3** Binding characteristics of anti-Env bispecific antibodies. **a** Schematic diagram of the bispecific binding assay via biolayer interferometry (BLI). **b** BLI response curves of bispecific binding assay. OCTET biosensors were loaded with biotinylated RSC3 (ligand 1) presenting the CD4bs epitope, and then probed sequentially with the bispecific antibody and BG505 SOSIP.664_D368R trimer (ligand 2) presenting the V3 glycan epitope. As controls, parental IgGs were used in place of the bispecific antibody. **c** Upper, negative stain EM of Bi-ScFv$_{dVRC01-5X-PGT121}$, in complex with BG505 SOSIP.664 Env at a molar ratio of 0.5:1 (second from left and middle panel) and 6:1 (second from right and right panel). The PGT121 and VRC01 variable regions are highlighted in green and purple, respectively (middle and right panel); Lower, schematic presentation of Env trimer and the epitopes for PGT121 (green) and VRC01 (purple), side view (left), and top view (right)

ΔE1I2/V3S (Fig. 2a) displayed the best neutralization breadth (90% virus coverage; Fig. 4a, Supplementary Fig. 3A-B).

Furthermore, the homogeneity of our lead bispecific antibodies, Bi-ScFv$_{dVRC01-5X-PGT121}$ and Bi-NAb$_{dVRC01-5X-PGT121}$, was verified by dynamic light scattering (DLS) and size exclusion chromatography (SEC) (Supplementary Fig. 4A-C). Both Bi-ScFv$_{dVRC01-5X-PGT121}$ and Bi-NAb$_{dVRC01-5X-PGT121}$ displayed unimodal distribution by DLS with z-average hydrodynamic diameters of 9.8 and 19.2 nm correspondingly. This was generally consistent with SEC data. A minor second peak, visible for Bi-NAb in SEC profile and size distribution by intensity in DLS, represented only 0.1% of the sample when calculated by volume (DLS), indicating negligible level of aggregation. In addition, we utilized BLI to determine the binding kinetics of dVRC01–5X-PGT121 ScFv with HIV Env BG505 SOSIP.664 trimer in

comparison with that of the parental ScFvs. We observed overall pico-molar affinity ($K_D < 1.0 \times 10^{-12}$ M), with similar low dissociation rate ($k_{off}$) below $1.0 \times 10^{-7}$ 1/s for all ScFvs to BG505 SOSIP.664 (Supplementary Fig. 4D & E). It is notable that both the parental ScFvs display remarkably low dissociation rate ($k_{off}$) for BG505 SOSIP.664 trimer, indistinguishable from that of Bi-ScFv$_{dVRC01-5X-PGT121}$, suggesting that Bi-ScFv bispecific binding mode has moderate contribution to the gain of avidity for typical HIV Env trimers such as BG505 for which parental bNAbs already have exceptional affinity.

Subsequently, we tested the top bispecific antibodies, Bi-ScFv$_{dVRC01-5X-PGT121}$ and Bi-NAb$_{dVRC01-5X-PGT121}$, against a more comprehensive panel of viruses covering the major genetic subtypes and circulating recombinant forms, and containing almost entirely of primary isolate Envs[45]. In this 208 virus panel,

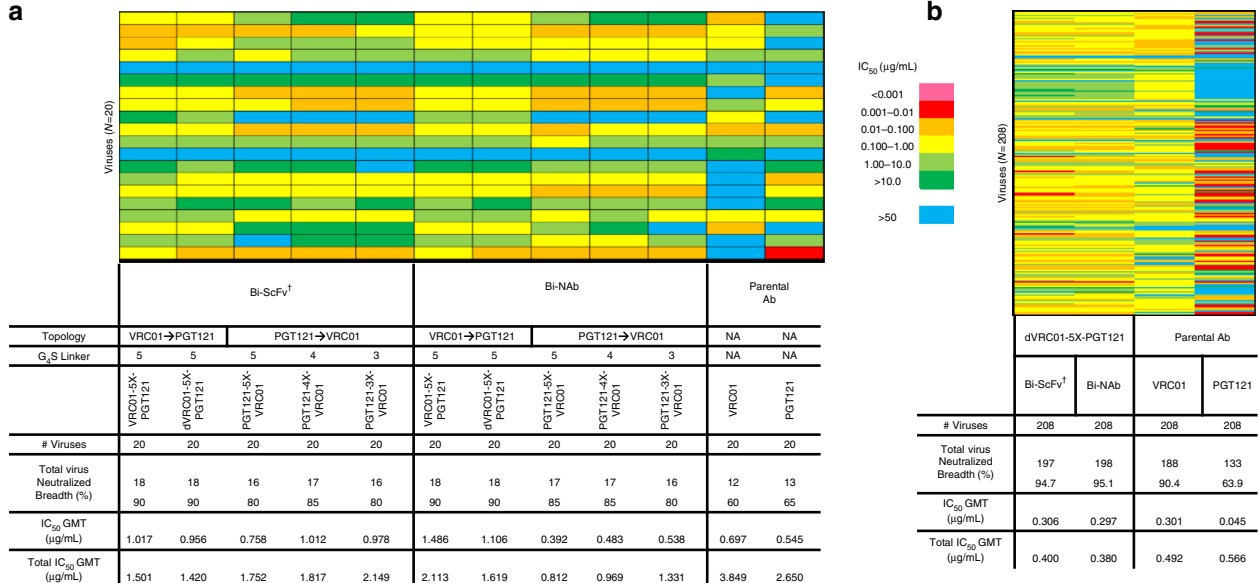

**Fig. 4** Neutralization breadth and potency of bispecific antibodies. **a** Neutralization breadth of the parental and bispecific antibodies was tested against an HIV-1 pseudovirus panel consisting of Envs of 20 viral strains. Heat maps of $IC_{50}$ titers were generated in Excel. In the heatmaps, each row represents a virus strain while columns represent antibodies. Warmer colors indicate more potent neutralization and blue indicates at 50 μg/mL, antibody virus neutralization is below detection threshold (see legend). Breadths based on $IC_{50}$s are also summarized. Potencies ($IC_{50}$ geometric mean values, GMT) were calculated against sensitive viruses. Total $IC_{50}$ GMT value, is also shown when $IC_{50}$ titers for all tested virus were included and an $IC_{50}$ value of 50 μg/mL was assigned to a neutralization-resistant virus ([†] indicates that the $IC_{50}$ was adjusted by a factor of 3 to account for the molarity difference between the lower molecular weight Bi-ScFv and the IgG or Bi-NAb). **b** Neutralization breadth of the parental and bispecific antibodies was tested against a panel of 208 viral strains. Heat maps of $IC_{50}$, breadth and potency are shown as in **a**

the Bi-ScFv$_{dVRC01-5X-PGT121}$ and Bi-NAb$_{dVRC01-5X-PGT121}$ displayed improved coverage with 94.7 and 95.1% of viruses neutralized, respectively, over the parental antibodies (VRC01 = 90.4%, and PGT121 = 64%; Figs. 4b and 5a) and with an overall potency ($IC_{50}$ geometric mean) comparable to VRC01 (Figs. 4b & 5a), consistent with the observation that Bi-ScFv$_{dVRC01-5X-PGT121}$ has high pico-molar affinity ($K_D$) and low dissociation rate ($k_{off}$) for BG505 SOSIP.664 trimer similar with the parental ScFvs (Supplementary Fig. 4D & E). Conversely, both Bi-ScFv$_{dVRC01-5X-PGT121}$ and Bi-NAb$_{dVRC01-5X-PGT121}$ were able to neutralize three primary isolates with dual resistance to individual or cocktail of parental bNAbs (Fig. 5a, b and Supplementary Fig. 3C & D), suggesting a cooperative effect exerted likely by the inter-protomer crosslinking of two distinct epitopes within the neutralization-resistant HIV-1 Env trimer. The gain of avidity through inter-protomer epitope crosslinking mode used by bispecific antibodies appears to be more prominent for Env trimers poorly neutralized by parental bNAbs.

**Generation and validation of a trispecific antibody.** Previous studies indicated that the HIV-1 gp41 MPER-specific bNAb, 10E8, when combined with gp120-specific bNAbs including the CD4bs bNAb VRC01, displayed an additive, and potentially small synergistic/cooperative effect on neutralization[46]. Here, we combined the MPER-specific 10E8 functional moiety with our top lead bispecific antibodies to improve neutralization breadth and potency further. We used knob-into-hole technology[32] to generate a heterodimeric Tri-NAb consisting of 10E8 and Bi-NAb$_{dVRC01-5X-PGT121}$ moieties (Supplementary Fig. 5A). The 10E8 was placed on one antibody arm with the Fc containing the "knob" mutation, and the Bi-ScFv$_{dVRC01-5X-PGT121}$ was fused to the IgG1 Fc containing the "hole" mutation on the other arm. To express the Tri-NAb, we co-transfected 293F cells with the plasmid DNA encoding the heavy and light chain genes of 10E8

and the Bi-NAb$_{dVRC01-5X-PGT121}$. We then purified the Tri-NAb via a protein A column and assessed its homogeneity by SDS–PAGE (Supplementary Fig. 5B).

We used BLI to assess the triple specificity of the Tri-NAb, with ligands presenting the epitopes of VRC01, PGT121, and 10E8 (Supplementary Fig. 5C & D), respectively. In a BLI Octet RED96 system, we loaded the ligands and antibodies to the biosensor surface sequentially in the following order: (1) biotinylated RSC3 as the initial ligand (ligand 1) to present VRC01 epitope; (2) the parental bNAbs or Tri-NAb; (3) the second ligand, trimeric BG505.SOSIP.664_D368R with CD4bs/VRC01 epitope knockout to present PGT121 epitope (ligand 2); and finally (4) the third ligand (ligand 3), an MPER peptide fused to a rabbit Fc (MPER rFc) (Supplementary Fig. 5C & D) to present the 10E8 epitope (Supplementary Fig. 5D). As expected, we found that Tri-NAb displayed binding signals for all of the three ligands, while the parental bNAbs only showed binding to ligand 1 with CD4bs epitope (e.g., VRC01) or no binding to any ligands (e.g., PGT121 and 10E8) due to the lack of initial CD4bs ligand 1 engagement in this sequential binding assay. Our data confirmed that the Tri-NAb containing the moieties of three bNAbs (VRC01, PGT121, and 10E8) could recognize all of the three cognate bNAb epitopes on HIV-1 Env molecule. We further analyzed the size profile of our Tri-NAb using SEC (Supplementary Fig. 5F) and DLS (Supplementary Fig. 5G & Supplementary Fig. 5H) and observed that similarly to parental antibody VRC01, the Tri-NAb is characterized with unimodal size distribution with a z-average hydrodynamic diameter of 12.8 nm.

**Elevated neutralization capacity of trispecific antibody.** We then assessed the neutralization capacity of the Tri-NAb against the comprehensive virus panel consisting of Envs of 208 HIV strains[45]. The Tri-NAb's neutralization capacity (indicated by $IC_{50}$ titers) was extraordinary as it neutralized all but one virus

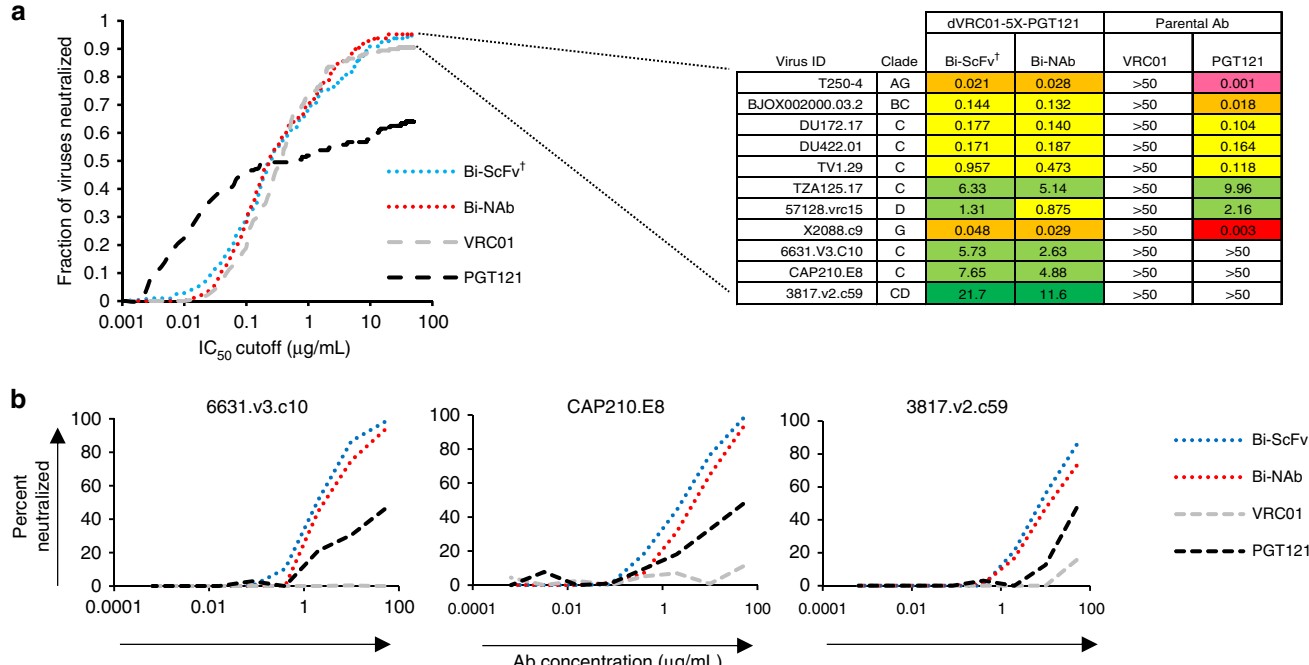

**Fig. 5** Bispecific antibody neutralization potency-breadth curves. **a** Potency-breadth curves comparing the bispecific antibodies to their parental IgGs in the neutralization assay with 208 viral strains (left panel) and summary of IC50 titers (µg/mL) against VRC01-resistant and dual-resistant viruses that are sensitive to the bispecific antibodies (right panel). ($^{†}$ indicates that the IC50 was adjusted by a factor of 3 to account for the molarity difference between the lower molecular weight Bi-ScFv and the IgG or Bi-NAb). **b** Raw neutralization curves of dual-resistant viruses sensitive to the Bi-ScFv and Bi-NAb in **a**

(99.5%) of the panel, thus showing much greater neutralization breadth than the parental bNAbs and the Bi-NAb $_{dVRC01-5X-PGT121}$ (Fig. 6a). In addition, the Tri-NAb neutralization potency was greater than that of the individual parental bNAbs or the Bi-NAb (Fig. 6b, ****$p < 0.0001$, Wilcoxon matched-pairs signed rank test), with an outstanding IC50 geometric mean of 0.069 µg/mL, which is at least four-fold lower than that of the VRC01, 10E8, and Bi-NAb$_{dVRC01-5X-PGT121}$ antibodies (Fig. 6a–c, Fig. 7a, Supplementary Fig. 6). The IC80 titers of the Tri-NAb corroborated its surpassing neutralization potency over both the parental bNAbs and the Bi-NAb$_{dVRC01-5X-PGT121}$ (Supplementary Fig. 5E, ****$p < 0.0001$, Wilcoxon matched-pairs signed rank test), neutralizing 98% of all strains with an IC80 geometric mean of 0.298 µg/mL (Supplementary Fig. 5E, Supplementary Fig. 7).

We attribute the gain in neutralization breadth exhibited by the Tri-NAb to the addition of 10E8 moiety to the Bi-NAb$_{dVRC01-5X-PGT121}$ agent (Fig. 6c). In this regard, several PGT121/VRC01 dual-resistant strains that are sensitive to 10E8 are also sensitive to Tri-NAb neutralization (Fig. 6c, right panel). The only virus strain that showed resistance to the Tri-NAb, 6471.V1.C16, is also highly resistant to both PGT121 and VRC01, and is only moderately sensitive to 10E8 (IC50 = 4.98 µg/mL; Supplementary Fig. 6). A bivalent 10E8 moiety such as that found in an IgG molecule may be required for neutralizing viruses moderately sensitive to 10E8, which is absent in our trispecific antibody (Supplementary Fig. 6).

To evaluate the impact that each antibody contributed to the Tri-NAb neutralization potency, we grouped viruses by their sensitivities to the parental antibodies and assessed the change in potency of neutralization of the multispecific NAbs from that of the parental antibodies (Fig. 7b). Bi-ScFv and Bi-NAb exhibited a 2.1-fold and 1.9-fold higher neutralization activity than VRC01 when measuring activity against VRC01/PGT121 dual sensitive viruses ($N = 125$) accounting for 60% of the 208 tested viruses. With viruses resistant to VRC01 and/or PGT121 ($N = 83$, 40% of

the tested viruses), the overall potency of the Bi-ScFv and Bi-NAb was lower than that of the parental bNAbs. To the end, when incorporating all viruses in the analysis, the potency of the bispecific antibodies was similar to that of VRC01 but lower than that of PGT121 (Fig. 7b, upper panel).

The Tri-NAb neutralized viruses sensitive to two antibody moieties ($N = 194$ or 93% of tested viruses) with greater potency than that of the parental antibodies (Fig. 7b, lower panel). However, when the viruses were only sensitive to one of the Tri-NAb's moieties (dual resistant, $N = 14$ or 6.7% of tested viruses), the potency was lower than that of the parental antibody (Fig. 7b, lower panel). The Tri-NAb displayed an overall higher potency than VRC01 and 10E8 by more than 4-fold for all viruses of the panel (Fig. 7b, lower panel). Therefore, the higher neutralization capacity of Tri-NAb than those of Bi-NAb and individual parental antibodies closely correlated with the Env binding avidity (targeting at least two epitopes) in the face of neutralization resistance.

## Discussion

Despite the improved breadth and potency displayed by bNAbs recently isolated, bNAb combinations will likely be necessary to cope with the extensive HIV Env diversity present in circulating virus strains. Recently some Bi-NAbs have been developed that combine receptor- or coreceptor-targeting antibodies or domains such as the anti CD4 receptor Fab ibalizumab (iMab)[47, 48], the anti CCR5 coreceptor Fab PRO140 (P140)[49], or the eCD4Ig containing a fusion protein of human CD4 ectodomain and a CCR5-mimetic sulfopeptide[50]. Although these antibody molecules can exhibit broad and potent HIV neutralization, they are different from the Bi- and Tri-NAb molecules described here, in that they target host receptors or use host receptor derivative, whereas our Bi- and Tri-NAb target only viral Env elements. A number of in vitro studies demonstrated that anti-HIV Env

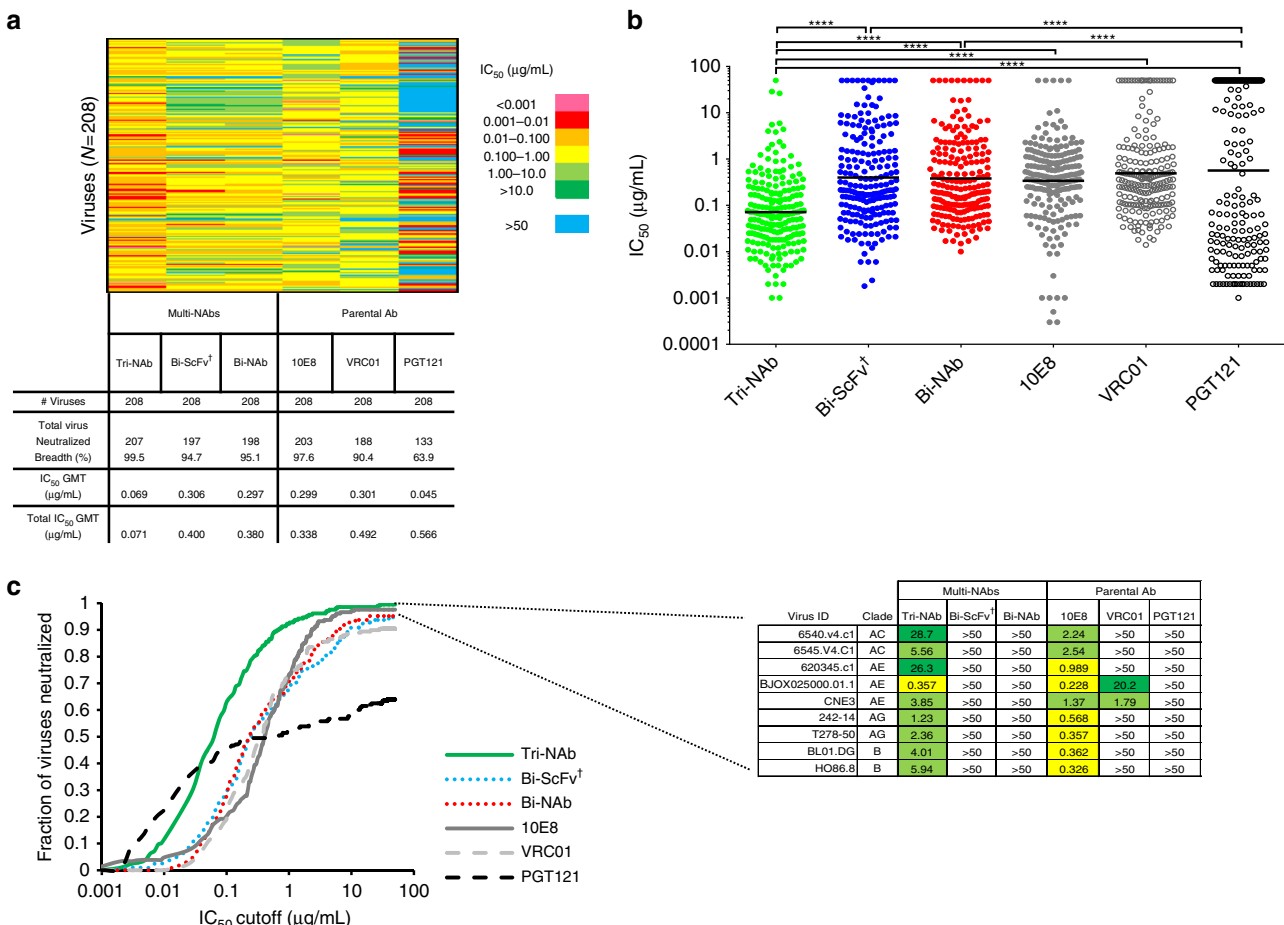

**Fig. 6** Neutralization breadth and potency of trispecific antibodies. **a** Neutralization breadth of the parental, bispecific and trispecific antibodies was tested against a panel of 208 viral strains. Heat maps of IC$_{50}$, breadth and potency are shown as in Fig. 4a. **b** Scatter plots of IC$_{50}$ titers in which each virus is represented by an individual dot. Total IC$_{50}$ GMT value is indicated by a black line for each antibody. Statistical differences in neutralization were evaluated using non-parametric *t*-test (Wilcoxon matched-pairs signed rank test) with *$p < 0.05$, **$p < 0.01$, ***$p < 0.001$, ****$p < 0.0001$. **c** Potency-breadth curves comparing the Tri-NAb to the Bi-NAb as well as the parental (left panel) and summary of IC$_{50}$ (μg/mL) against viruses that are resistant to Bi-NAb but sensitive to the Tri-NAb (right panel). $^{†}$ indicates that the IC$_{50}$ was adjusted by a factor of 3 to account for the molarity difference between the lower molecular weight Bi-ScFv and the IgG or Bi-NAb

bNAb combinations yield superior neutralization results[45, 46]. This is likely due to the cooperative effect when two bNAbs targeting independent epitopes on the Env trimer are combined. Therefore, combining three or four bNAbs with different epitope specificities may be of significant therapeutic value[46, 51]. Empirical combinations of bNAbs can achieve enhanced potency and breadth over parental bNAbs alone[31, 34, 35] and it has been suggested that some may be capable of crosslinking the Env trimer[31, 34]. Specifically, one study utilized DNA-linkers as "molecular rulers" to construct molecules capable of intra-spike crosslinking with the ultimate goal of enhancing the potency of bNAbs[31] by gain of avidity.

In this study, we utilized structure-based antibody rational design to optimize the multi-epitope engagement of the HIV Env trimer with Bi-ScFvs antibodies. We validated both the dual engagement of Env epitopes and the enhanced neutralization capacity of the Bi-ScFvs. Furthermore, we confirmed that our Bi-NAb engages the CD4bs and the V3 base glycan epitopes through an inter-protomer linkage rather within one Env trimer following our structure-based design (Figs. 1 and 2, Supplementary Table 1).

Neutralization analyses of the Bi-NAbs using a panel of viruses with differential resistance profiles illustrated that those possessing both the VRC01 VL at the N-terminus of the bispecific

antibody as well as the VRC01 VL N-terminal mutations (ΔE1I2/V3S) had improved neutralization breadth (Fig. 4a, Supplementary Fig. 3A-B). Conversely, we observed that Bi-NAbs with PGT121 at the N-terminus had improved potency but reduced neutralization breadth, suggesting that this molecular configuration may present unfavorable steric constraints. Furthermore, we observed that molecules with shorter linker lengths of 4X or 3X displayed lower neutralization breadth and potency than those with longer (5X) linkers (Figs. 4a, b and 5a), suggesting that longer linkers may enable the bispecific antibodies to orchestrate individual functional moieties for simultaneously engaging both cognate epitopes.

We combined the Bi-NAb and MPER-specific bNAb 10E8 to form a Tri-NAb with profoundly improved neutralization breadth (Fig. 6a–c, Supplementary Fig. 5E, Supplementary Fig. 6 & Supplementary Fig. 7). This gain of breadth most likely attributed to the addition of the 10E8 moiety, as PGT121/VRC01 dual-resistant viral strains that are sensitive to 10E8 were sensitive to Tri-NAb neutralization (Fig. 6c). Consistently, the potency of the Tri-NAb (IC$_{50}$ geomean of 0.069 μg/mL, IC$_{80}$ geometric mean of 0.298 μg/mL) increased in comparison with the Bi-NAb (Fig. 6a) and the parental antibodies (Figs. 6a–c, Supplementary Fig. 5E).

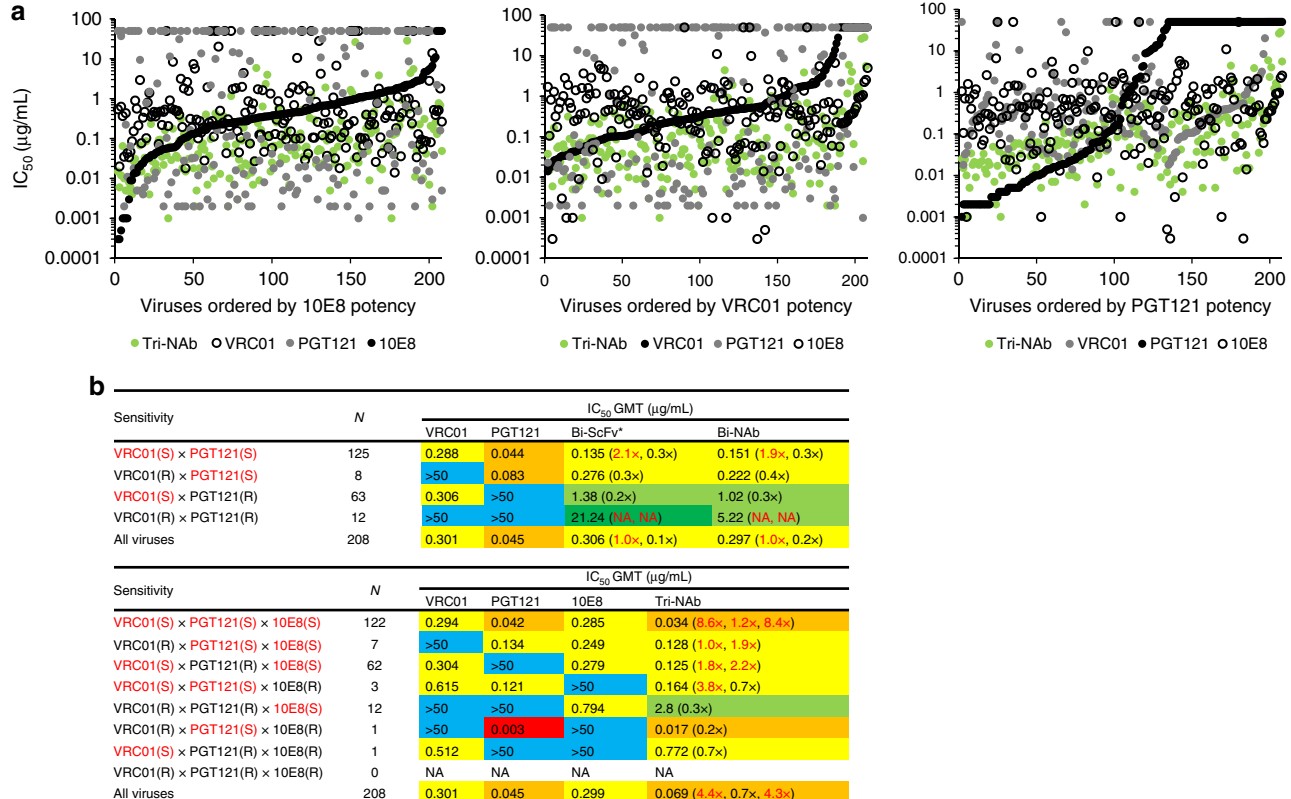

**Fig. 7** Analysis of multispecific antibody neutralization breadth and potency. **a** Comparison of Tri-NAb potency to parental IgGs potencies. Each dot on the graph represents a virus plotted in decreasing order of sensitivity to the reference parental IgG (black). The potency of the Tri-NAb (green), the second parental IgG (gray), and the third parental IgG (open circle) against the same virus is overlaid with data points below the parental IgG indicating increased potency. **b** Neutralization potency (IC$_{50}$ geometric mean values, GMT) of Bi-NAb (upper panel) and Tri-NAb (lower panel) compared to the parental antibodies. Viruses were grouped by parental antibody sensitivity, with S indicating sensitive and R, resistant. N, the number of viruses under each sensitivity group. IC$_{50}$ GMT titer fold change reflecting potency compared to the respective antibodies is indicated in the parentheses next to the multispecifc antibody potency. Fold changes in red indicating improved potency. ($^\dagger$ indicates that the IC$_{50}$ was adjusted by a factor of 3 to account for the molarity difference between the lower molecular weight Bi-ScFv and the IgG, Bi-NAb or Tri-NAb)

Both Bi- and Tri-NAb outperformed their parental mAbs in neutralization potency when tested with viral isolates that were sensitive to all of the individual bNAbs (Fig. 7b); most likely due to the cooperative effect resulted from simultaneous engagement of multiple epitopes within the same Env trimer. Consistently, the Tri-NAb neutralizes these sensitive viruses more potently than the Bi-NAb (Fig. 7b). The cooperative effect remains for the Tri-NAb when tested with viruses resistant to only one antibody (single-resistance; Fig. 7b), while this cooperative effect is lost for the Bi-NAb (Fig. 7b). When dual neutralization-resistant viruses were tested, the Tri-NAb showed steady neutralization with slightly lower potency compared to the parental mAbs (Fig. 7b), while the Bi-NAb only occasionally neutralized the viruses with moderate potency (Fig. 7b). The profound improvement of neutralization of the Tri-NAb over Bi-NAb against both sensitive and single-resistant viruses, which represent the majority of the global circulating viruses, strongly highlights the premise of engineering multi-epitope (more than two epitopes) targeting antibodies. We anticipate that the incorporation of additional bNAb functional moieties to the Tri-NAb will improve its neutralization capacity even further.

We successfully combined three HIV-1 broadly neutralizing antibody specificities into one single trispecific antibody by structure-based rational design. The triple-specificity antibody demonstrated outstanding HIV viral coverage (99.5%) in neutralization assays of a 208 virus panel with remarkable potency

(IC$_{50}$ GMT below 0.1 µg/mL). With these outstanding neutralization capacities, the Tri-NAb represents an excellent candidate for the next generation of HIV-1 preventive and therapeutic antibody-based agents.

In the neutralization assay using a virus panel consisting 208 virus isolates, the Tri-NAb missed one highly resistant virus strain, 6471.V1.C16 (Supplementary Fig. 6), which can be neutralized by one of the parental bNAbs, 10E8. This suggests that a combination of multispecific antibody with selected complimentary monoclonal antibody may achieve superior efficacy to conventional antibody cocktail or multispecific antibody alone. This unconventional format of connecting multiple antibody ScFvs by flexible linkers, however, may result in potential in vivo immunogenicity. Therefore, in vivo pharmacokinetics and immunogenicity studies of this triple-specific antibody in experimental animals such as human neonatal Fc receptor (FcRn) knock in mice[52] or non-human primates are required to assure the feasibility of further applications. In a recent study by Xu et al.[53] using knob-into-hole heterodimerization and CVDVIg configuration to combine three HIV bNAb specificities resulted in trispecific antibodies with neutralization breadth similar to the Tri-NAb generated in this study. By choosing parental bNAbs PGDM1400 (V1V2 glycan specific)[54] and N6 (CD4bs specific)[55] that are more potent than PGT121 and VRC01, respectively, the trispecific antibody N6/PGDM1400-10E8v4 by Xu et al.[53] has three-fold higher potency than the Tri-NAb in this study, suggesting that selection of parental bNAbs developed more recently with

outstanding neutralization profiles[54–57] could substantially advance the antiviral capacity of multispecific antibodies. Furthermore, various Fc mutations, especially the "LS" mutations (M428L/N434S) conferring enhanced affinity for neonatal Fc receptor that improves the in vivo half-life and biodistribution of antibody molecules could be incorporated into the multispecific antibody context to potentially improve the utility of passive immunization[44, 58, 59]. Coordinating combinations of additional antibody entities into multi-NAb designs will become more feasible, and may further the development of strategies for HIV-1 infection prevention, remission, and possible eradication in the future.

## Methods

**Analysis of Ab structures in complex with SOSIP.** The published structure of the two bNAbs PGT122 and VRC01 bound to JR-FL SOSIP.664 (PDB: 5FYK)[40] was used to model bispecific combinations of VRC01 and PGT121 in the program Chimera[41] where PGT122 served as the surrogate for PGT121. Trimers were visualized using the Higher Order Structure-Unit Cell tool. Antibody variable domains were retained for analysis, while antibody residues corresponding to the CL1 and CH1 regions were deleted from the Fab entities. Distances between termini were assessed by selecting the respective terminis' Cβ atoms using the Structure Analysis-Distance tool and measuring the spatial distance.

**Antibody production.** Bi-NAbs in a ScFv format were designed utilizing tetra-glycine-serine (G$_4$S) peptide linkers[36, 37]. C-terminal His-tagged Bi-ScFvs DNA sequences were synthesized (GenScript, Piscataway, NJ) and cloned into the pcDNA3.1(-) vector (Thermo Fisher Scientific) while the Bi-ScFv lacking the C-terminal His-tag was subcloned into an IgG1 Fc vector (InvivoGen) as well as IgG1 Fc vectors carrying knob-into-hole mutations[32]. Bi-NAbs were expressed by transient transfection of either Bi-ScFv or Bi-ScFv-IgG1 Fc plasmids in FreeStyle 293 F cells (Thermo Fisher Scientific, Catalog number: R79007). Tri-NAbs were expressed by transient transfection of the Bi-ScFv$_{dVRC01-5X-PGT121}$ IgG1 Fc "knob", 10E8 HC "hole" and 10E8 LC plasmids in 293 F cells. Supernatants were harvested 5 days post transfection, filtrated, followed by affinity purification. Bi-ScFvs with 6-His-tag were purified by Complete His-tag purification resin (Sigma-Aldrich). Bi-NAb and Tri-NAb with IgG1 Fc were purified by protein A affinity chromatography. Elutes were dialyzed with phosphate-buffered saline (PBS), and concentrated using an Amicon Ultra 10 kDa molecular weight cutoff concentrator (Sigma-Aldrich). Antibody purity was analyzed by SDS–PAGE.

**Size exclusion chromatography.** Size exclusion chromatography was carried out to resolve protein size profiles and remove potential aggregates and undesired oligomeric forms. 300–500 µl of protein with a concentration ranging from 1–2 mg/mL was injected to Superose 6 10/300GL column with 5–5000 kDa fractionation range (GE Healthcare) pre-equilibrated with PBS, followed by elution with PBS at 0.5 mL/minute in an ÄKTA Pure Station (GE Healthcare).

**Dynamic light scattering (DLS).** Dynamic light scattering (DLS) was performed using a Zetasizer Nano series, ZEN3600 (Malvern Instruments Ltd., Worcestershire, UK) and analyzed using Malvern Zetasizer 7.10 software (Malvern Instruments Ltd., Worcestershire, UK). Samples were prepared in PBS, pH 7.4 at a concentration of 1 mg/mL and filtered using Millex 0.22 µm filters (Millipore Sigma) prior to the analysis.

**Protein purification.** Avi-tagged BG505 SOSIP.664[60] WT and D368R, and RSC3 core[8] were expressed in 293 F cells, purified using a *Galanthus nivalis* (GN)-lectin agarose (Vector Laboratories) column as previously described[61], followed by purifications with size-exclusive chromatography (SEC) with purity confirmed by Blue NativePage. The BirA 500 biotin ligase (Avidity Avitag™ Technology) was utilized according to the manufacturer's protocol to biotinylate the C-terminal avi-tags of BG505 SOSIP.664 and RSC3.

**Ab binding affinity.** Biolayer light interferometry (BLI) was performed using an Octet RED96 instrument (ForteBio; Pall Life Sciences). Bispecific binding was confirmed by first capturing biotin labeled RSC3 (ligand 1) at 10 µg/mL onto Streptavidin biosensors for 300 s. The biosensors were then submerged in binding buffer (PBS/0.2% TWEEN 20) for a wash for 60 s followed by immersion in a solution containing 250 nM of either the parental or Bi-NAb antibodies for 180 s and an immediate immersion in a solution containing 300 µg/mL of trimeric BG505 SOSIP.664_D368R (ligand 2) for 300 s.

Trispecific binding was confirmed by first capturing 10 µg/mL of biotin labeled RSC3 (ligand 1) onto Streptavidin biosensors (ForteBio; Pall Life Sciences) for 300 s. The biosensors were then submerged in binding buffer for a wash of 60 s. Then, the biosensors were immersed in a solution containing 250 nM of either the parental or Bi-NAb antibodies for 180 s followed by an immediate immersion in a solution containing 300 µg/mL of trimeric BG505 SOSIP.664_D368R (ligand 2) for 300 s. Finally, the biosensors were immersed in a solution containing 50 µg/mL of MPER peptide fused to a rabbit Fc (MPER rFc) for 120 s. Baselines were established before and after the loading step. All assays were performed in 1 × binding buffer.

To evaluate the affinity of bi-ScFv and parental ScFvs for BG505 SOSIP.664 trimer, biotinylated BG505 SOSIP.664 trimer was captured by Streptavidin biosensors in binding buffer at 10 µg/mL followed by baseline wash and immersed into wells containing ScFv in 2-fold dilution series starting at 500 nM. $K_D$, the affinity constant value in molars was calculated as $k_{off}/k_{on}$. The sensograms were corrected with the blank reference and fit with the software ForteBio Data Analysis 9.0 using a 1:1 binding model with the global fitting function (grouped by color, Rmax).

**HIV-1 neutralization assays.** Ab neutralization assays were performed in a single round of infection using HIV-1 Env pseudoviruses and TZM-bl target cells (NIH AIDS Reagent Program, Catalog number: 8129), as previously described[62, 63]. Neutralization curves were fitted by nonlinear regression using a five-parameter hill slope equation as previously described[62]. The $IC_{50}$ or $IC_{80}$ titers of Abs were reported as the concentration of Ab required to inhibit infection by 50 and 80%, respectively. The $IC_{50}$ or $IC_{80}$ geometric mean (Geomean) indicating mAb neutralization potency was derived from $IC_{50}$ or $IC_{80}$ values against each individual tier 2 virus for each mAb. When $IC_{50}$ or $IC_{80}$ value is >50 µg/mL for certain viruses (no neutralization), a value of 50 µg/mL is designated for calculation. The number of viruses neutralized by mAb ($IC_{50}$ or $IC_{80}$ < 50 µg/mL) out of the total number of tested viruses was used to calculate the neutralization breadth. The virus panel for the neutralization profile covers the major genetic subtypes and circulating recombinant forms and consists almost entirely of primary isolate Envs[45].

**Electron microscopy analysis.** To confirm bivalent binding, Bi-ScFv/BG505 SOSIP.664 complexes were generated by incubating 6× molar ScFv with BG505 SOSIP.664 overnight at room temperature, followed by purification of complexes by SEC. To specifically confirm crosslinking of the HIV Env trimer, ScFv/BG505 SOSIP.664 complexes were generated by incubating 0.5× molar ScFv with BG505 SOSIP.664 overnight at room temperature, followed by purification of complexes by SEC.

Complexes were then deposited on 400 mesh copper grids and stained with 2% uranyl formate. Negative stain EM images were taken on a 120 kv Tecnai Spirit microscope with a LAB6 filament. Raw micrographs were collected using Leginon[64] and deposited in Appion[65]. DoG picker[66] was performed to select particles in stain. Those particles were stacked and aligned using Iterative MSA/MRA[67]. 2D classes representing the complex are shown in Fig. 3 and Supplementary Fig. 2. Images were false colored using Photoshop.

**Statistical analysis.** Comparisons of antibody neutralization performance was carried out with one-way ANOVA. Statistical evaluation of difference between two groups was performed via non-parametric *t*-test for paired data, determined with Wilcoxon matched-pairs signed rank test, with a two-tailed *p* value calculated for significance. Statistical significance was determined as *$p$ < 0.05, **$p$ < 0.01, ***$p$ < 0.001, ****$p$ < 0.0001. All statistical analysis was performed with GraphPad Prism version 7.

**Data availability.** The authors declare that the data supporting the findings of this study are available within the article and its Supplementary Information files, or are available from the authors upon request.

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

## Acknowledgements

This work is supported by Intramural Research Award from the Institute for Bioscience and Biotechnology Research, University of Maryland, and Maryland Innovation Initiative (2017-MII-3992) to Y.L. This work is also partially supported by the Bill and Melinda Gates Foundation CAVD grant OPP1115782 to A.B.W., and funding from the Intramural Research Program of the Vaccine Research Center, National Institute of Allergy and Infectious Diseases, NIH. J.S. is a trainee of NIH training grant T32AI125186A to Anne Simon at University of Maryland, College Park.

## Author contributions

Conceptualization: Y.L., J.J.S., and J.G.; Methodology: J.J.S., M.L., A.K.A., and R.T.B.; Investigation: J.J.S., H.L.T., K.M., M.L., S.O., C.C., L.L., A.G., and N.R.; Visualization: J.J. S., J.G., and H.L.T.; Supervision: Y.L., J.R.M., A.B.W., R.T.B., A.K.A., and N.R; Project Administration: Y.L.; Funding Acquisition: Y.L., A.B.W., J.R.M., and R.T.B.; Writing—Original Draft: Y.L. and J.J.S.; Writing—Review & Editing: Y.L., J.J.S., J.G., H.L.T., A.B. W., and J.R.M.

## Additional information

**Competing interests:** The authors declare no competing interests.

