## [Peer Review File · Nature Communications]

Reviewers' comments:

Reviewer #1 (Remarks to the Author):

This study addresses the interesting question whether a single engineered antibody-like agent is capable of simultaneously targeting distinct HIV Env epitopes resulting in improved HIV neutralizing potency and breadth. Based on structural analysis of the interactions of HIV Env trimer with bNAbs, the authors generated a panel of Bi-ScFvs. Using negative stain EM, they showed that optimal Bi-ScFv is capable of crosslinking adjacent protomers within one HIV Env spike. However, when testing viral neutralizing activity in vitro, optimal Bi-ScFv improves only the neutralizing breadth, but not the potency (to any significance), over its parental bNAbs. In addition, the authors combined this bispecific antibody with a third bNAb 10E8 moiety and showed that a tri-specific antibody-like molecule further improves neutralizing breadth and potency in vitro. Based on these results, authors concluded that a combination of multi-functional moieties of individual bNAbs can elevate antibody binding avidity, hence improving HIV neutralizing potency and breadth.

& #x00A0; &# x00A0;

There are several important limitations that raise questions as to whether the data provide insight into antibody bispecific binding, i.e., bivalent binding, binding avidity and neutralizing potency. Previously, Asokan and Mascola showed that bispecific antibodies (generated via CrossMab technology) capable of targeting two epitopes on Env improved neutralizing breadth when compared to their respective single parental bNAbs, but the overall breadth and potency was similar to the combination of two parental bNAbs. The bispecific antibodies' failure to improve neutralizing potency is likely due to the limited reach between two Fabs. Using the "molecular ruler" strategy, Galimidi and Bjorkman nicely showed that intra-spike crosslinking can indeed enhance the binding avidity and importantly improve the potency of bNAbs. Furthermore, Bournazos and Ravetch showed that compared to the combination of two parental antibodies, a bispecific antibody design based on a longer IgG3 hinge can not only significantly improve the neutralizing potency and breadth, but also enhance anti-HIV activity in humanized mouse model. These studies showed consistently that engineered bispecific antibodies can improve neutralizing breadth due to the contribution from each Fab arm. However, improving avidity requires crosslinking two epitopes, and the binding avidity is expected to correlate with enhanced neutralizing potency. The strength of the current study is that authors utilized structure-based antibody rational design to optimize the multi-epitope engagement of Env trimer with Bi-ScFv antibodies. But the optimally designed bispecific antibodies with dual engagement of Env epitope, validated by negative stain EM, can only improve breadth but failed to enhance virus neutralization potency to any significant degree. Scientifically, the reason for this discrepancy is not well addressed. Avoiding such discussion present a major weakness of the manuscript.

The authors should have compared the neutralization results of their bispecific and trispecific antibodies with the neutralization profiles of the particular double and triple combinations of the parental antibodies. If there is any simultaneous binding of two or three epitopes, the potency of the multispecific antibody should be superior. This is yet another major deficiency of the study. Furthermore, the authors do not directly compare their

results to those generated by Pace et al, Sun et al, Gardner et al, Bournazos et al and Huang et al. Based on an “eyeball” examination, the constructs being reported here are significantly inferior in neutralization potency. Therefore, as potential HIV prevention products, they face superior competition.

Lastly, if there is no significant new science being revealed, then the constructs must have practical application as HIV preventive agents. However, there is little evidence that these constructs could be developed into real products. Given the described antibody structures differ significantly from a normal antibody structure, there is real concerns regarding their stability, PK properties, and potential immunogenicity, especially given the linker sequences inserted. More in vitro analyses would be helpful, and preliminary in vivo determination of PK and anti-HIV activity would be informative. But they are completely missing.

Specific comments:

1. Given the weaknesses of this study outlined above, the current title is an overstatement and should be modified to reflect the limitation of the rational design strategy. The potency is certainly far from exceptional.
2. As stated already, when comparing the neutralizing breadth and potency of bispecific moieties, the data from the combination of two parental antibodies should be used. This is to provide meaningful comparisons.
3. On line 177, authors stated “SEC with negligible aggregation forms (Fig. S1B)”. However, the SEC data presented suggests a significant portion of small shoulder peak was presented. Since the detailed SEC method was not presented, it is very difficult to discern if this small shoulder peak fraction (which seems to have multiple small peaks) was an aggregation. Detailed SEC method should be provided. A control molecule (monoclonal antibody that provide a single peak) should also be provided.
4. Reduced SDS-PAGE analysis is not an appropriate methodology to measure the homogeneity of the antibody. Non-Reduced SDS-PAGE analysis and data should be included.
5. SEC data for trispecific antibody should be presented to demonstrate the developability of this molecule.
6. Data presented in figure 3 C and D were interesting. It would be more informative if, for three dual resistant viruses (6631, CAP210 and 3817), the neutralization curve for bispecific antibodies, parental bNAbs and combination parental bNAbs against individual virus were presented. ;
7. With respect to future development, in addition to the positive aspects of the bi- and tri-specific antibodies, the authors should also discuss the limitations of such irregular antibody-like structures, for example, potential immunogenicity, developability concerns, etc. The manuscript could be more useful to the field if they could show in vivo activity of their constructs, say in a humanized mouse model of HIV infection.

Reviewer #2 (Remarks to the Author):

This is a straightforward but interesting bit of protein engineering, in which two scFvs are

strung together with an Fc by (G4S)_n linkers and then paired with the heavy chain of a third antibody. As expected, when all three variable regions recognize epitopes that are targets of bnAbs, the potency and breadth are both extremely high. While the work does not break any new conceptual ground, it is probably interesting enough as a concrete application of previously validated ideas to merit, after suitable editing, publication in a journal like Nature Communications.

Suitable editing. The overall result is expected, if the engineering is OK, and the data speak for themselves, although it is, of course, nice to see that it works (and useful also). So the first step is to remove the entire Discussion, which is pure repetition, keeping only the last paragraph (which might be called Conclusions).

Since the MS suffers a bit from "template writing" and some hints of writing by a non-native speaker, the next steps are outlined below.

line 29 We describe here joining single-chain [You didn't do the joining "here" -- you are only describing the joining "here". You did the joining in the lab.]

line 33 has greater neutralization breadth than its parental bnAbs [use the 3-letter "has" rather than the more pretentious 'demonstrates'; "superior" is a value judgment, and scientific prose should be objective; the correct comparative in English is "greater ... than", not "over" or "compared to" or anything else]

line 35 which has nearly pan-isolate neutralization breadth and high potency [the sentence as written is simply ungrammatical English]

line 65 ... viruses, but these viruses have much lower genetic diversity than do circulating HIV-1 isolates. [see remark about correct English comparative above]

line 69 The HIV rapidly develops resistance under pressure from a single bnAb, however, suggesting that passive treatment [fixing awkward syntax]

lines 80-82 Preliminary data on bnAb cocktails suggest substantial advantages over either cART or single bnAb treatments for the management of HIV-1 infection. [you have an advantage "over", not "compared to" -- also word order needs fixing, as shown]

line 87 Their neutralization breadth could be further extended, however, and truly ["improved" is a value judgment; use an objective term like "extended"; avoid "however" at the beginning of a sentence, unless you mean "in whatever way" -- as "However nasty and pedantic this reviewer might be, I still find his critique instructive."]

line 100 delete "however" (not needed)

line 102 We therefore sought to expand

line 112 We show here that

line 139 delete "interestingly" -- let the reader decide whether it is interesting

lines 143 and 144: 37.2 Å and 53.5 Å Citing distances measured on a model to five significant figures is the kind of mistake expected of a high-school freshman.

line 150 delete "herein" [wrong usage]

line 154 delete "briefly" [makes no sense here]

line 156 delete "namely 3-5X linkers" [makes no sense]

line 157. The empirical choice of 3-5 G4S linkers was to avoid potential steric hindrance imposed by [sentence needed rewriting to make sense]

lines 163 and 165 wrong use of "topology" -- "sequence" is fine in both places and in a later place also

line 174 "in total" [delete "a" for correct English]

line 175 "ran as homogeneous species" (there are various Abs, so "species" here is plural)

line 179 "We used ..." [why waste space on a longer and uglier synonym?]

line 180 The Env ligands included RSC3 core,

line 186 could bind ["could" = "were able to" and costs you only one word, not three]

lines 206-7 given the sub-sturating ratio of Bi-ScFv and Env trimer. We found [let the reader decide what's important]

line 207 was decorated with only one of the [change word order]

line 218 can fully occupy the total of 6 cognate epitopes

lines 236 and 238 use "sequence" not "topology"

line 280 can recognize (instead of "is capable of recognizing")

line 286 ... panel, thus showing much greater breadth than the parental

line 288 ... was greater than that of [stop pushing value judgments on me -- the facts, man, just the facts]

line 296 We attribute the gain to the addition

line 307 2.1-fold and 1.9-fold higher neutralization activity than VRC01

line 311 the overall potency of the Bi-ScFv and of the Bi-nAb was lower than that of the ...

line 313 ... 1.4 and 1.3 fold greater than that of [as I wrote above, the correct English comparative takes "than" -- "compared to" is actually nonsense here]

line 315 delete "of note"

line 316 with greater potency [delete "a"]

line 330 an overall greater poeency than VRC01

line 321 Therefore, the higher neutralization capacity of Tri-nAb than those of Bi-nAb and individual parental Abs closely correlated with the Env avidity [the thought is restating the obvious, making the overall logic almost circular, but at least use correct English to express it]

.....

line 387 the Tri-nAb is an excellent candidate for the next

Why not use Fig. S2 instead of Fig 2C -- show real data, not some hand-picked single image

Reviewer #3 (Remarks to the Author):

The field of therapeutic antibody formats is quickly developing especially for infectious diseases as so much good science has shown the need to target the pathogen at multiple epitopes in order to weaken the pathogen's ability to adapt by mutation. Furthermore, the demonstration that passive immunization, provided by an antibody format, provokes additional mechanisms of ultimate protection such as strengthening the adaptive and endogenous humoral responses, makes the exploration of optimal configurations essential for success. Steinhardt and colleagues have taken a rationale design approach to the problem. Their study starts with a simple construct and builds to a higher and higher avidity through more complicated antibody-based structures leading to increasingly significant capacity to neutralize across a very broad panel of viral strains. The explanations are well thought through and presented with the following comments:

The rationale for choices of target epitopes are validated with figure 1 providing a clear explanation of this as well as for the design of the constructs. The demonstration of the co-binding is convincing, with beautiful visual support provided by the electron micrographs of the binding interfaces using the subsaturating concentration group.

Generally, as I am not an expert on the details of the viral neutralization assays, the data looks convincing. However, I challenge the helpfulness of providing a GEO mean in figure 3A

and B as, for example in B, the BiSc-Fv has several red lines while the Bi-Nab format has only one red line, yet they have equivalent Geo Means (0.306 vs 0.297). The way the data is presented in figure 4 B is a better way to communicate the message.

Staying on figure 3, the data presented in that figure are used by the authors to state in the results and conclusion that the linker length is relevant, i.e., '...molecules with shorter linker lengths of 4x or 3x displayed lower neutralization breadth...' This is not true as the breadth data presented in fig 3A show 80/85/80 or 85/85/80 % viruses neutralized for biScFv vs Bi-Nab for 5x/4x/3x formats, respectively. In other words, 4x is actually always the best. So either the data is wrong in the table or I am not able to read the table correctly. So please address perhaps by adding the numbers for breadth in the text of the results or changing this statement.

Furthermore, it is not clear how the data of table D of figure 3 is related to the curves of figure 3C. For example, the values for the Parental Ab with VRC01 are all <50 yet it's line on the graph is similar to the lines of the other two parameters in the graph but in the table to the left, they have much different values. (And also, where is the D panel noted in the legend of figure 3?)

The data for the trispecific versus the other two formats in figure 4 are well presented and convincing.

The main concern that I have with this manuscript is the lack of better presentation of the analytics to prove/convince the reader of the homogeneity of the biAb or TriAb batches. The reason to know more accurately the level of aggregates is that in this biological system, having multimers may falsely increase the neutralization capacity, even if the level of aggregates is 'relatively' low. We know this from our own experience. Thus, it would be necessary to provide information on the size range of the column used. Also, please provide light scattering info to confirm the mass of the peaks as I don't agree that what is marked as the aggregate peak in fig S1 is actually that. There are peaks closer to the main peaks that should be where the aggregate peaks are. Furthermore, the trispecific structure is by the nature of putting all these Ab fragments together, a manufacturing challenge. Thus, having more analytical info than what is provided in fig S4 is essential to be able to validate the declaration of low aggregates/homogeneity to support the biological results that is the structure of a bi or tri Ab and not the aggregates.

One minor comment is to fix the sentence in the Intro on line 79 as there seems not to be needed 'as well as'.

This study and publication of the manuscript are very timely and will be of genuine interest to the field. Just this week, Science has published two articles in this domain, highlighting the importance to medicine:

'Why is the flu vaccine so mediocre?' J Cohen Science 22 Sep 2017. 'There's an increasing call now to improve the vaccine and organize the research community to more collaboratively try to develop a "universal" flu shot that works against many strains and lasts for many years, if not a lifetime.'

Trispecific broadly neutralizing HIV antibodies mediate potent SHIV protection in macaques Ling Xu1 et al., Science 20 Sep 2017:

The development of an effective AIDS vaccine has been challenging due to viral genetic diversity and the difficulty in generating broadly neutralizing antibodies (bnAbs). Here, we engineered trispecific antibodies (Abs) that allow a single molecule to interact with three independent HIV-1 envelope determinants: 1) the CD4 binding site, 2) the membrane

proximal external region (MPER) and 3) the V1V2 glycan site. Trispecific Abs exhibited higher potency and breadth than any previously described single bnAb, showed pharmacokinetics similar to human bnAbs, and conferred complete immunity against a mixture of SHIVs in non-human primates (NHP) in contrast to single bnAbs. Trispecific Abs thus constitute a platform to engage multiple therapeutic targets through a single protein, and could be applicable for diverse diseases, including infections, cancer and autoimmunity. I would suggest that as the authors see that sometimes the biAb or TriAb do not do as well as the parental mAb for certain viral strains, they might comment in the discussion on how yet a future approach could do even better (e.g., mixtures). But that suggestion is optional. Thus, post addressing the above comments, my opinion would be favorable for publication of this manuscript in Nature Communications.

Point-by-Point Response to Reviewer's Comments

We appreciate the overall positive and constructive comments from the reviewers. After carefully revised our manuscript, with the edited text highlighted, we are providing point-by-point response to their comments as below.

Reviewer #1

Comments

This study addresses the interesting question whether a single engineered antibody-like agent is capable of simultaneously targeting distinct HIV Env epitopes resulting in improved HIV neutralizing potency and breadth. Based on structural analysis of the interactions of HIV Env trimer with bNAbs, the authors generated a panel of Bi-ScFvs. Using negative stain EM, they showed that optimal Bi-ScFv is capable of crosslinking adjacent protomers within one HIV Env spike. However, when testing viral neutralizing activity in vitro, optimal Bi-ScFv improves only the neutralizing breadth, but not the potency (to any significance), over its parental bNAbs. In addition, the authors combined this bispecific antibody with a third bNAb 10E8 moiety and showed that a tri-specific antibody-like molecule further improves neutralizing breadth and potency in vitro. Based on these results, authors concluded that a combination of multi-functional moieties of individual bNAbs can elevate antibody binding avidity, hence improving HIV neutralizing potency and breadth.

There are several important limitations that raise questions as to whether the data provide insight into antibody bispecific binding, i.e., bivalent binding, binding avidity and neutralizing potency. Previously, Asokan and Mascola showed that bispecific antibodies (generated via CrossMab technology) capable of targeting two epitopes on Env improved neutralizing breadth when compared to their respective single parental bNAbs, but the overall breadth and potency was similar to the combination of two parental bNAbs. The bispecific antibodies' failure to improve neutralizing potency is likely due to the limited reach between two Fabs. Using the "molecular ruler" strategy, Galimidi and Bjorkman nicely showed that intra-spike crosslinking can indeed enhance the binding avidity and importantly improve the potency of bNAbs. Furthermore, Bournazos and Ravetch showed that compared to the combination of two parental antibodies, a bispecific antibody design based on a longer IgG3 hinge can not only significantly improve the neutralizing potency and breadth, but also enhance anti-HIV activity in humanized mouse model. These studies showed consistently that engineered bispecific antibodies can improve neutralizing breadth due to the contribution from each Fab arm. However, improving avidity requires crosslinking two epitopes, and the binding avidity is expected to correlate with enhanced neutralizing potency. The strength of the current study is that authors utilized structure-based antibody rational design to optimize the multi-epitope engagement of Env trimer with Bi-ScFv antibodies. But the optimally designed bispecific antibodies with dual engagement of Env epitope, validated by negative stain EM, can only improve breadth but failed to enhance virus neutralization potency to any significant degree. Scientifically, the reason for this discrepancy is not well addressed. Avoiding such discussion present a major weakness of the manuscript.

Response: We appreciate the reviewer's insight to the mechanism regarding the contrast effect of bispecific antibody on the improvement of neutralization breadth and potency: great gain of breadth but moderate improvement in potency. We believe that the gain of breadth is mainly caused by the combination of two parental bNAb moieties in the intra-spike protomer crosslinking mode used by the bispecific antibody.

The similarity of potency between that of the bispecific antibody and the parental bNAbs may be caused by a number of factors: 1) steric hindrance imposed by the Env structural elements from diverse HIV isolates which are not reflected by the current available structural information; and 2) the parental antibodies, VRC01 and PGT121 displaying **remarkably low dissociation rates** (slower than 10^{-7} 1/s) (**provided as new data in Fig. S4 D & E**) to the Env trimers such as BG505 SOSIP.664, which is virtually identical to the bispecific scFv. This observation is consistent with an elegant study by Galimidi et al. (2015, Cell 160, 433–446) revealing that for off-rates slower than 10^{-5} 1/s, avidity enhancement by multivalence binding is not substantial. Therefore, in this case, we do not see substantial enhancement of potency possessed by our bispecific antibody with many virus isolates. We addressed this point Perhaps a combination of other parental antibodies with higher off-rate for trimer will lead to more prominent gain of potency.

However, when tested with **dual resistant** viruses, the bispecific antibodies display greater potency than the individual parental antibodies (**Fig. 3C & D**) as well as a cocktail of the parental antibodies (**Fig. S3C & D**). In these cases, both parental Abs likely have poor affinity (possibly a fast off-rate as well) for the resistant viral Env spikes and fail to neutralize the viruses, whereas the bispecific antibody could overcome the low affinity by bivalent binding to the Env spike and mediate neutralization.

Comments

The authors should have compared the neutralization results of their bispecific and trispecific antibodies with the neutralization profiles of the particular double and triple combinations of the parental antibodies. If there is any simultaneous binding of two or three epitopes, the potency of the multispecific antibody should be superior. This is yet another major deficiency of the study.

Response

We had long-standing neutralization assay experience with antibody combinations in the lab (Mascola lab). We had tested more than 200 antibodies and antibody derivatives and compared their anti-viral capacities. The accumulated data suggest that, overall, the improvement of neutralization breadth by antibody combination was closely predicted by an additive-effect model and explained by complementary neutralization profiles of antibodies recognizing distinct epitopes (Kong et al., J Virol. 2015 Mar 1; 89(5): 2659–2671). Thus, the parental antibody neutralization profile as a side-by-side control used in this study is sufficient to show the effect of physical combination of bNAbs on neutralization breadth and potency.

Nevertheless, we performed a confirmatory neutralization assay in which the bispecific antibodies, parental antibodies, and a cocktail of parental antibodies were included to compare the breadth and potency of the antibodies against a VRC01-resistant virus panel (**Fig. S3C & D**). The outcome of this examination is highly consistent with our previous data with large virus panel (208-virus panel) (**Fig. 3C &**

D). Therefore, we are confident with our large virus panel neutralization data presented in this manuscript.

Comments

Furthermore, the authors do not directly compare their results to those generated by Pace et al, Sun et al, Gardner et al, Bournazos et al and Huang et al. Based on an “eyeball” examination, the constructs being reported here are significantly inferior in neutralization potency. Therefore, as potential HIV prevention products, they face superior competition.

Response

It is a good point to have a comparison of these anti-viral agents. However, due to the difference between the design rationale (different target) and the various virus neutralization assay panels used by these different studies, it is practically difficult to compare them fairly.

For instance, as stated in the introduction section of the manuscript, the studies by Pace et al, Sun et al, and Huang et al targeted host factor and one Env epitope, while the anti-viral agent in the study of Gardner et al is based on receptor CD4 (host factor). In contrast, our trispecific antibody targets three epitopes on viral Env. We believe that our unique multi-epitope targeting trispecific antibody is a good addition to these innovative agents. Indeed, it will be interesting to combine these approaches to generate novel anti-viral agents in future studies.

The Bournazos et al study created a bispecific antibody targeting CD4bs and V3 glycan with 93% virus coverage (IC50 GMT 0.04 ug/ml), while our bispecific and trispecific antibody displays 95% (IC50 GMT = 0.3 ug/ml) and 99.5% virus coverage (IC50 GMT= 0.07 ug/ml), respectively. Again, it is hard to compare them, as the assay virus panel is different between these studies. Nevertheless, our approach generated antibodies with “comparable” antiviral potency and greater breadth. In addition, our trispecific antibody targets three Env epitopes, which has greater potential to combat viral resistance than bispecific antibodies. Furthermore, the cross-spike binding mode conveyed by rational design based tandem scFVs in this study will be of general interest for the field to engineer novel antibodies.

Comments

Lastly, if there is no significant new science being revealed, then the constructs must have practical application as HIV preventive agents. However, there is little evidence that these constructs could be developed into real products. Given the described antibody structures differ significantly from a normal antibody structure, there is real concerns regarding their stability, PK properties, and potential immunogenicity, especially given the linker sequences inserted. More in vitro analyses would be helpful, and preliminary in vivo determination of PK and anti-HIV activity would be informative. But they are completely missing.

Response

We appreciate the reviewer’s suggestion for investigations toward practical applications including the in vivo PK, immunogenicity, and anti-HIV activity, although they are beyond the scope of our current study.

We are planning to perform such experiments in the future after we publish our work here and secure more fund to support the suggested studies.

Specific comments:

1. *Given the weaknesses of this study outlined above, the current title is an overstatement and should be modified to reflect the limitation of the rational design strategy. The potency is certainly far from exceptional.*

Response: There are two critical components of antiviral capacity: breadth and potency. Our Tri-NAb displays 99.5% virus coverage, which is exceptional among all the solely Env-targeting antibodies. Regarding the potency, our Tri-NAb achieves IC₅₀ GMT around 0.07 ug/ml, quite comparable to the other solely Env-targeting antibodies.

We revised the title of our manuscript to “Rational Design of a Trispecific Antibody Targeting the HIV-1 Env with Outstanding Anti-viral Activity”

2. *As stated already, when comparing the neutralizing breadth and potency of bispecific moieties, the data from the combination of two parental antibodies should be used. This is to provide meaningful comparisons.*

Response

As stated previously, we performed a confirmatory neutralization assay in which the bispecific antibodies, parental antibodies, and a cocktail of parental antibodies were included to compare the breadth and potency of the antibodies against a VRC01-resistant virus panel (**Fig. S3C & D**). The outcome of this examination is highly consistent with our previous data with large virus panel (208-virus panel) (**Fig. 3C & D**). The effect of antibody combination on neutralization breadth and potency was closely predicted by an additive-effect model and explained by complementary neutralization profiles of antibodies recognizing distinct epitopes. Therefore, we are confident with our large virus panel neutralization data presented in this manuscript without testing all the 208 viruses with the combination of two parental antibodies. The data of parental antibodies as control is sufficient to serve as control.

3. *On line 177, authors stated “SEC with negligible aggregation forms (Fig. S1B)”. However, the SEC data presented suggests a significant portion of small shoulder peak was presented. Since the detailed SEC method was not presented, it is very difficult to discern if this small shoulder peak fraction (which seems to have multiple small peaks) was an aggregation. Detailed SEC method should be provided. A control molecule (monoclonal antibody that provide a single peak) should also be provided.*

Response

We appreciate that the reviewer pointed out the interpretation of the SEC data. We recognized the “shoulder peak” in the SEC and revised the text accordingly. We provided the details of the SEC in the methods section, and added VRC01 as a control in the SEC (**Fig. S4A**). In addition, we performed DLS (**Fig. S4A-C**) and showed the details of the molecular sizes of the bispecific antibodies (**Lines 240-246**).

4. *Reduced SDS-PAGE analysis is not an appropriate methodology to measure the homogeneity of the antibody. Non-Reduced SDS-PAGE analysis and data should be included.*

Response

We added the Non-Reduced SDS-PAGE analysis data in **FIG. S1 and S5B**.

5. *SEC data for trispecific antibody should be presented to demonstrate the developability of this molecule.*

Response

We added the SEC data for trispecific antibody (**Fig. S5F-H, Lines 298-301**).

6. *Data presented in figure 3 C and D were interesting. It would be more informative if, for three dual resistant viruses (6631, CAP210 and 3817), the neutralization curve for bispecific antibodies, parental bNAbs and combination parental bNAbs against individual virus were presented.*

Response

We added the data in **Fig. 3D**, and added an independent confirmatory data set in **Fig. S3C & D**.

7. *With respect to future development, in addition to the positive aspects of the bi- and tri-specific antibodies, the authors should also discuss the limitations of such irregular antibody-like structures, for example, potential immunogenicity, developability concerns, etc. The manuscript could be more useful to the field if they could show in vivo activity of their constructs, say in a humanized mouse model of HIV infection.*

Response

We revised our manuscript and addressed these points (**Lines 414-419**). As discussed previously, we are planning to perform such in vivo efficacy experiments in the future after we publish our work here and secure more fund to support the suggested studies.

Reviewer #2

Comments

This is a straightforward but interesting bit of protein engineering, in which two scFvs are strung together with an Fc by (G4S)_n linkers and then paired with the heavy chain of a third antibody. As expected, when all three variable regions recognize epitopes that are targets of bNAbs, the potency and breadth are both extremely high. While the work does not break any new conceptual ground, it is probably interesting enough as a concrete application of previously validated ideas to merit, after suitable editing, publication in a journal like Nature Communications.

Suitable editing. The overall result is expected, if the engineering is OK, and the data speak for themselves, although it is, of course, nice to see that it works (and useful also). So the first step is to remove the entire Discussion, which is pure repetition, keeping only the last paragraph (which might be called Conclusions).

Response

We appreciate the encouragement of the reviewer and we are very thankful for the truly constructive and instructive suggestions to improve the manuscript. We made most of the revision accordingly.

Comments

Since the MS suffers a bit from "template writing" and some hints of writing by a non-native speaker, the next steps are outlined below.

line 29 We describe here joining single-chain [You didn't do the joining "here" -- you are only describing the joining "here". You did the joining in the lab.]

line 33 has greater neutralization breadth than its parental bnAbs [use the 3-letter "has" rather than the more pretentious "demonstrates"; "superior" is a value judgment, and scientific prose should be objective; the correct comparative in English is "greater ... than", not "over" or "compared to" or anything else]

line 35 which has nearly pan-isolate neutralization breadth and high potency [the sentence as written is simply ungrammatical English]

line 65 ... viruses, but these viruses have much lower genetic diversity than do circulating HIV-1 isolates. [see remark about correct English comparative above]

line 69 The HIV rapidly developes resistance under pressure from a single bnAb, however, suggesting that passive treatment [fixing awkward syntax]

lines 80-82 Preliminary data on bnAb cocktails suggest substantial advantages over either cART or single bnAb treatments for the management of HIV-1 infection. [you have an advantage "over", not "compared to" -- also word order needs fixing, as shown]

line 87 Their neutralization breadth could be further extended, however, and truly ["improved" is a value judgment; use an objective term like "extended"; avoid "however" at the beginning of a sentence, unless you mean "in whatever way" -- as "However nasty and pedantic this reviewer might be, I still find his critique instructive."]

line 100 delete "however" (not needed)

line 102 We therefore sought to expand

line 112 We show here that

line 139 delete "interestingly" -- let the reader decide whether it is interesting

lines 143 and 144: 37.2 Å and 53.5 Å Citing distances measured on a model to five significant figures is the kind of mistake expected of a high-school freshman.

line 150 delete "herein" [wrong usage]

line 154 delete "briefly" [makes no sense here]

line 156 delete "namely 3-5X linkers" [makes no sense]

*line 157. The empirical choice of 3-5 G4S linkers was to avoid potential steric hindrance imposed by
[sentence needed rewriting to make sense]*

Response: All the points listed above were corrected.

Comment

lines 163 and 165 wrong use of "topology" -- "sequence" is fine in both places and in a later place also

Response:

We thank the reviewer for making the suggestion. We think that "topology" describes the configuration of the antibody molecule well, however. The other reviewers are fine with this term and use. Therefore, we will remain using "topology" to refer to the sequence order of the antibody functional moieties.

Comments

line 174 "in total" [delete "a" for correct English]

line 175 "ran as homogeneous species" (there are various Abs, so "species" here is plural)

line 179 "We used ..." [why waste space on a longer and uglier synonym?]

line 180 The Env ligands included RSC3 core,

line 186 could bind ["could" = "were able to" and costs you only one word, not three]

lines 206-7 given the sub-sturating ratio of Bi-ScFv and Env trimer. We found [let the reader decide what's important]

line 207 was decorated with only one of the [change word order]

line 218 can fully occupy the total of 6 cognate epitopes

lines 236 and 238 use "sequence" not "topology"

line 280 can recognize (instead of "is capable of recognizing")

line 286 ... panel, thus showing much greater breadth than the parental

line 288 ... was greater than that of [stop pushing value judgments on me -- the facts, man, just the facts]

line 296 We attribute the gain to the addition

line 307 2.1-fold and 1.9-fold higher neutralization activity than VRC01

line 311 the overall potency of the Bi-ScFv and of the Bi-nAb was lower than that of the ...

line 313 ... 1.4 and 1.3 fold greater than that of [as I wrote above, the correct English comparative takes "than" -- "compared to" is actually nonsense here]

line 315 delete "of note"

line 316 with greater potency [delete "a"]

line 330 an overall greater poetrncy than VRC01

line 321 Therefore, the higher neutralization capacity of Tri-nAb than those of Bi-nAb and individual parental Abs closely correlated with the Env avidity [the thought is restating the obvious, making the overall logic almost circular, but at least use correct English to express it]

line 387 the Tri-nAb is an excellent candidate for the next

Response: All the points listed above were corrected.

Comment

Why not use Fig. S2 instead of Fig 2C -- show real data, not some hand-picked single image

Response: We think that the images in Fig. 2C makes the point well that the bispecific scFv crosslinks the adjacent protomers within one trimer. For more details, the reader can look at more images in Fig. S2. Indeed, the images in Fig. 2C represent those in Fig. S2 well.

Reviewer #3

Comments

The field of therapeutic antibody formats is quickly developing especially for infectious diseases as so much good science has shown the need to target the pathogen at multiple epitopes in order to weaken the pathogen's ability to adapt by mutation. Furthermore, the demonstration that passive immunization, provided by an antibody format, provokes additional mechanisms of ultimate protection such as strengthening the adaptive and endogenous humoral responses, makes the exploration of optimal configurations essential for success. Steinhardt and colleagues have taken a rationale design approach to the problem. Their study starts with a simple construct and builds to a higher and higher avidity through more complicated antibody-based structures leading to increasingly significant capacity to neutralize across a very broad panel of viral strains. The explanations are well thought through and presented with the following comments:

The rationale for choices of target epitopes are validated with figure 1 providing a clear explanation of this as well as for the design of the constructs. The demonstration of the co-binding is convincing, with beautiful visual support provided by the electron micrographs of the binding interfaces using the subsaturating concentration group.

Response: We appreciate the enthusiasm of the reviewer.

Comments

Generally, as I am not an expert on the details of the viral neutralization assays, the data looks convincing. However, I challenge the helpfulness of providing a GEO mean in figure 3A and B as, for example in B, the BiSc-Fv has several red lines while the Bi-Nab format has only one red line, yet they have equivalent Geo Means (0.306 vs 0.297). The way the data is presented in figure 4 B is a better way to communicate the message.

Response: We thank the reviewer for looking at the data carefully and pointed that we should have done a better job to clarify the data. The original IC50 Geo mean presented in Fig. 3A and B was calculated using IC 50 values only against SENSATIVE viruses. Therefore, PGT121, with many “blue” strips (resistant, therefore not counted), displays a much lower IC50 Geo mean than the other antibodies with better breadth.

To clarify this point, in the current **Fig. 3A and B**, we renamed the original IC50 Geo mean as “IC50 GMT” (Geometric titer), which was derived from SENSATIVE virus IC50 titers. The field uses IC50 Geomean to describe the potency against SENSATIVE viruses, which is the “working” concentration of antibodies. In addition, we used another term, “Total C50 GMT” to describe IC 50 when all IC 50 values including those for the resistant viruses. In the case of resistant viruses, an IC 50 value of 50 ug/ml (the highest antibody concentration in the neutralization assay) was assigned to the corresponding antibody. In another word, IC50 GMT describes the potency against the Sensitive viruses, which reflects the effective working concentration of particular antibody; whereas Total IC50 GMT describes the overall potency of a given antibody with the consideration of breadth (since IC50 values against resistant viruses are included). For example, PGT121 has 63% breadth with many resistant viruses also has very low IC50 GMT but high Total IC50 GMT. For antibodies such as the Bi-NAb with higher breadth than PGT121, the IC50 GMT and Total IC50 GMT is very close. We think that the two forms of IC50 GMT values would help to better describe the neutralization breadth and potency of a given antibody.

Comments

Staying on figure 3, the data presented in that figure are used by the authors to state in the results and conclusion that the linker length is relevant, i.e., ‘...molecules with shorter linker lengths of 4x or 3x displayed lower neutralization breadth...’ This is not true as the breadth data presented in fig 3A show 80/85/80 or 85/85/80 % viruses neutralized for biScFv vs Bi-Nab for 5x/4x/3x formats, respectively. In other words, 4x is actually always the best. So either the data is wrong in the table or I am not able to read the table correctly. So please address perhaps by adding the numbers for breadth in the text of the results or changing this statement.

Response: We thank the reviewer for spotting this inconsistency in the text. We corrected the text in **lines 234-236**, reads as “ We also noted that the five-or four-tandem G4S linker length (5X or 4X) was

slightly better than the shorter variants (3X) (Fig. 3A, S3A-B) in many cases”.

Comments

Furthermore, it is not clear how the data of table D of figure 3 is related to the curves of figure 3C. For example, the values for the Parental Ab with VRC01 are all <50 yet it's line on the graph is similar to the lines of the other two parameters in the graph but in the table to the left, they have much different values. (And also, where is the D panel noted in the legend of figure 3?)

Response: The Table D is now the right panel of the current Fig. 3C. This is to show the gain of breadth from VRC01 (90% virus coverage) to the bispecific antibody (95% coverage) in the 208 virus panel assay. The gain of 5% breadth looks small in the P-B curve on the left panel. The right panel is a detailed description about how the 5% breadth gain occurs, showing the VRC01 resistant virus isolates (N=11) can be neutralized by the Bi-NAb possessing the complimentary PGT121 moiety. Three dual resistant viruses are also sensitive to the Bi-NAb, suggesting cooperative effects of these two distinct moieties from parental antibodies. We now merged the legend text for the original panel C and D.

Comments

The data for the trispecific versus the other two formats in figure 4 are well presented and convincing. The main concern that I have with this manuscript is the lack of better presentation of the analytics to prove/convince the reader of the homogeneity of the biAb or TriAb batches. The reason to know more accurately the level of aggregates is that in this biological system, having multimers may falsely increase the neutralization capacity, even if the level of aggregates is 'relatively' low. We know this from our own experience. Thus, it would be necessary to provide information on the size range of the column used. Also, please provide light scattering info to confirm the mass of the peaks as I don't agree that what is marked as the aggregate peak in fig S1 is actually that. There are peaks closer to the main peaks that should be where the aggregate peaks are.

Furthermore, the trispecific structure is by the nature of putting all these Ab fragments together, a manufacturing challenge. Thus, having more analytical info than what is provided in fig S4 is essential to be able to validate the declaration of low aggregates/homogeneity to support the biological results that is the structure of a bi or tri Ab and not the aggregates.

Response: We provided additional SEC and DLS data to address the homogeneity of the biAb or TriAb batches in **Fig. S4A-C**, and **Fig. S5 F-H**, respectively, and described in text (**lines 240-247, 298-301**) accordingly. Briefly, the Bi-ScFv is highly homogenous in size, while Bi-NAb displays a minor aggregate peak in size distribution by intensity in DLS, which is virtually undetectable in size distribution by volume plot in DLS (account for 0.1% of the total particle population), indicating practically negligible level of aggregation. The Tri-NAb shows a single peak in DLS with no aggregate form, similar to the parental antibody VRC01.

Comments

One minor comment is to fix the sentence in the Intro on line 79 as there seems not to be needed 'as well as'

Response: Corrected.

Comments

This study and publication of the manuscript are very timely and will be of genuine interest to the field. Just this week, Science has published two articles in this domain, highlighting the importance to medicine:

'Why is the flu vaccine so mediocre?' J Cohen Science 22 Sep 2017. 'There's an increasing call now to improve the vaccine and organize the research community to more collaboratively try to develop a "universal" flu shot that works against many strains and lasts for many years, if not a lifetime.'

Trispecific broadly neutralizing HIV antibodies mediate potent SHIV protection in macaques

Ling Xu1 et al., Science 20 Sep 2017:

The development of an effective AIDS vaccine has been challenging due to viral genetic diversity and the difficulty in generating broadly neutralizing antibodies (bnAbs). Here, we engineered trispecific antibodies (Abs) that allow a single molecule to interact with three independent HIV-1 envelope determinants: 1) the CD4 binding site, 2) the membrane proximal external region (MPER) and 3) the V1V2 glycan site. Trispecific Abs exhibited higher potency and breadth than any previously described single bnAb, showed pharmacokinetics similar to human bnAbs, and conferred complete immunity against a mixture of SHIVs in non-human primates (NHP) in contrast to single bnAbs. Trispecific Abs thus constitute a platform to engage multiple therapeutic targets through a single protein, and could be applicable for diverse diseases, including infections, cancer and autoimmunity.

I would suggest that as the authors see that sometimes the biAb or TriAb do not do as well as the parental mAb for certain viral strains, they might comment in the discussion on how yet a future approach could do even better (e.g., mixtures). But that suggestion is optional.

Response: we took this notion and added the text **in lines 410-414**, reads, "In the neutralization assay using a virus panel consisting 208 virus isolates, the Tri-NAb missed one highly resistant virus strain, 6471.V1.C16 (**Fig. S7**), which can be neutralized by one of the parental bNAbs, 10E8. This suggests that a combination of multi-specific antibody with selected complimentary monoclonal antibody may achieve superior efficacy to conventional antibody cocktail or multi-specific antibody alone."

Comment

Thus, post addressing the above comments, my opinion would be favorable for publication of this manuscript in Nature Communications.

Response: We thank the reviewer for the supportive comments.

REVIEWERS' COMMENTS:

Reviewer #3 (Remarks to the Author):

The authors have addressed all my concerns satisfactorily. Thank you as much easier to evaluate and form an opinion with the data presented in this new manner. I have no further items to communicate. I find the data strong and their interpretation accurate. The study and its data will add significant value to the field and hopefully bring an effective therapeutic strategy to patients.

Reviewer #4 (Remarks to the Author):

This is an interesting study by Steinhardt et al. describing a new tri-specific reagent capable of neutralizing HIV. While it cannot be considered groundbreaking given the publication of numerous other multi-valent proteins against HIV, some with greater clinical potential than the one described here, the work is well executed and does add enough to what is known as to warrant publication. With respect to the comments by Reviewer #1, they raise valid criticisms and questions regarding the lack of improved potency for the tri-specific reagent. These questions are largely answered by the responses by the authors to the critique. It is conceivable that the specific choice of linker designs could limit any possible gains in potency that might be observed by other published configurations and the field should be made aware of this. The reviewer's point regarding the comparison to neutralization assays with combinations of single agents is a reasonable one to raise, but I agree with the authors that it has been shown fairly conclusively that combinations of single agents display additive (rather than synergistic) properties. For good measure, the authors have performed the requested assays and confirmed this for the combination of two antibodies. The last point made by the reviewer regarding the inferiority of this particular tri-specific reagent to others is accurate based on the available data, but I don't feel should preclude publication given that this could be another tool in the arsenal of reagents that have been developed against HIV. In my opinion, it is important for there to be second and third-line reagents in the pipeline in case the primary candidates fail during development.

Reviewers' comments after the revision

Reviewer #3

Comments

The authors have addressed all my concerns satisfactorily. Thank you as much easier to evaluate and form an opinion with the data presented in this new manner. I have no further items to communicate. I find the data strong and their interpretation accurate. The study and its data will add significant value to the field and hopefully bring an effective therapeutic strategy to patients.

Response: We agree with the reviewer and thanks for his/her comments.

Reviewer #4 (The journal was unable to obtain a report from Reviewer #1 and therefore recruited Reviewer #4 with similar expertise to comment on her/his concerns)

Comments

This is an interesting study by Steinhardt et al. describing a new tri-specific reagent capable of neutralizing HIV. While it cannot be considered groundbreaking given the publication of numerous other multi-valent proteins against HIV, some with greater clinical potential than the one described here, the work is well executed and does add enough to what is known as to warrant publication. With respect to the comments by Reviewer #1, they raise valid criticisms and questions regarding the lack of improved potency for the tri-specific reagent. These questions are largely answered by the responses by the authors to the critique. It is conceivable that the specific choice of linker designs could limit any possible gains in potency that might be observed by other published configurations and the field should be made aware of this. The reviewer's point regarding the comparison to neutralization assays with combinations of single agents is a reasonable one to raise, but I agree with the authors that it has been shown fairly conclusively that combinations of single agents display additive (rather than synergistic) properties. For good measure, the authors have performed the requested assays and confirmed this for the combination of two antibodies. The last point made by the reviewer regarding the inferiority of this particular tri-specific reagent to others is accurate based on the available data, but I don't feel should preclude publication given that this could be another tool in the arsenal of reagents that have been developed against HIV. In my opinion, it is important for there to be second and third-line reagents in the pipeline in case the primary candidates fail during development.

Response: We agree with the reviewer and thanks for his/her comments.